# Infinite Hope: Reframing Disconnection in Emerging Adulthood Through Purpose, Agency, and Identity

**DOI:** 10.3390/bs15091205

**Published:** 2025-09-04

**Authors:** William Terrell Danley

**Affiliations:** Division of Workforce Development and Lifelong Learning, University of the District of Columbia, Washington, DC 20032, USA; william.danleyjr@udc.edu

**Keywords:** agency, disconnected youth, emerging adults, hope theory, infinite mindset, Simon Sinek, youth re-engagement

## Abstract

Infinite Hope (IH) is a conceptual framework designed to restore identity, direction, and resilience among disconnected emerging adults. Integrating Snyder’s Hope Theory with Sinek’s Infinite Mindset, this paper examines how cultivating an infinite mindset strengthens agency and pathways, how alignment with a just cause mediates sustained goal pursuit, and under what conditions re-engagement flourishes. The model draws on an interdisciplinary review of seventy peer-reviewed sources and grounds its propositions in established sociological perspectives on agency, purpose, collective capacity, and human capabilities. As a conceptual and theoretical paper, it contributes a novel integration of psychological and sociological insights, addressing gaps in existing models that often overlook the interaction of hope, purpose, and collective environments. IH combines the hope triad with existential flexibility, courageous leadership, and values-based alignment, offering a testable framework that links psychological growth with social context. Visual models clarify their developmental sequence, and a research agenda outlines strategies for empirical validation. IH provides a practical blueprint for embedding meaning, reinforcing identity, and cultivating environments that sustain purposeful growth for youth-serving organizations.

## 1. Introduction

### 1.1. The Landscape of Disconnection

Connection plays a pivotal role in the lives of emerging adults, typically defined as individuals between the ages of 18 and 25 ([4]). This life stage involves the exploration of identity, making significant life decisions, and gradually assuming adult responsibilities, all of which require strong and sustained ties to supportive environments. Connection includes participation in education, employment, or civic life, alongside emotional bonds to peers, mentors, and institutions that provide structure, affirmation, and purpose ([10]; [33]). Research shows that such engagement builds self-confidence, psychological resilience, and a future-oriented mindset ([23]; [38]), enabling emerging adults to navigate uncertainty with direction. These connections form a protective framework supporting sustained growth when anchored in personal meaning and collective well-being.

Systemic and contextual factors profoundly shape the ability of emerging adults to form these connections. Structural inequalities in education, labor markets, housing, and healthcare disproportionately limit access for marginalized groups ([10]; [12]). National estimates indicate that over four million young people in the United States disconnect from education, employment, or structured training programs ([12]). Disconnection is not merely an individual shortcoming; it reflects disruptions in developmental processes caused by economic marginalization, under-resourced schools, inequitable mental health services, and unstable housing ([8]; [38]; [60]). These systemic barriers intensify stress and diminish the capacity to envision a coherent future, contributing to cycles of instability.

Moving beyond general accounts of barriers requires consideration of place-based and network processes that foster hope and reconnection. Collective efficacy and social capital provide a useful lens for understanding how community-level trust and shared expectations contribute to the formation of hope. [46] ([46]) demonstrate that when neighborhoods or communities cultivate norms of reciprocity and collaboration, they create conditions that mirror the “trusting teams” emphasized within the Infinite Mindset. These networks reinforce accountability and provide tangible resources that expand the capacity of emerging adults to re-engage.

Addressing these barriers requires the Infinite Hope (IH) framework to integrate targeted strategies beyond mindset cultivation and hope building. These strategies include expanding access to wraparound supports, fostering equitable hiring pipelines, partnering with community-based organizations to deliver mentorship, and advocating for policy reforms that address structural inequities. Such interventions create consistency in the context of developmental needs and ecological support, ensuring that reconnection efforts tackle the root causes of disengagement.

Even so, disconnection does not erase resilience or aspiration. Many emerging adults persevere with vision and determination to reconnect with opportunity, often despite longstanding systemic barriers ([7]; [28]). As [23] ([23]) note, conventional success metrics often fail to capture these internal strengths. A more expansive developmental framework must recognize both the structural challenges and the latent potential for reconnection.

### 1.2. Disconnection as a Developmental Interruption

Disconnection during emerging adulthood disrupts critical developmental tasks such as identity formation, long-term planning, and integrating cultural and community values into life goals ([9]; [29]). Without sustained engagement in education, employment, or community life, emerging adults experience emotional stagnation, diminished agency, and difficulty linking current actions to future aspirations. Structural inequities such as discriminatory hiring practices, neighborhood disinvestment, and limited access to culturally relevant mentorship compound these effects.

Identity processes are not fixed but involve ongoing internal conversations where young people reflect on possibilities, weigh competing demands, and reframe their identities in response to changing circumstances. Reflexivity provides a mechanism for these internal dialogues, helping emerging adults assess available options and adaptively reorient their goals when disruptions occur ([3]). Identity salience further shapes how these choices are prioritized, with certain roles or commitments exerting greater influence on action ([55]). When disconnection interrupts these processes, the capacity to reestablish coherence depends on individual resilience and supportive contexts.

For example, a young adult who leaves school due to financial strain may initially identify as disconnected and uncertain about future prospects. Through mentorship and structured opportunities, they may reflect on their experiences, reframe their identity around resilience, and reconnect with career training consistent with their values. This process illustrates how reflexivity and identity coherence can convert disruption into adaptive reorientation.

Viewing disconnection as a developmental interruption underscores the need for interventions that cultivate agency, belonging, and sustainable opportunities. Effective programs integrate cultural relevance, community leadership, and policy advocacy to move beyond compliance-based reintegration toward transformative engagement ([35]; [54]). Within the IH framework, identity coherence becomes central: courageous leadership emerges as value-guided action that links reflexive self-assessment with purposeful engagement. This integration positions emerging adults to transform developmental interruptions into renewed growth and sustained connection opportunities. The framework includes place-based programs connecting emerging adults to local networks, skills training reflecting labor market realities, and community initiatives reinforcing social capital ([29]; [37]).

### 1.3. Hope and the Infinite Mindset as Pathways to Re-Engagement

Building on this foundation, this article introduces a novel integration of Snyder’s Hope Theory and Simon Sinek’s Infinite Mindset to better understand how disconnected emerging adults can reconnect with purpose, direction, and possibility. Although both frameworks have shaped their respective fields, scholars have not yet synthesized them to inform re-engagement strategies for youth navigating systemic barriers. This conceptual model bridges that gap by combining motivational psychology with adaptive leadership philosophy, offering unique advantages over alternative approaches.

Alternative motivational models, such as [45]’s ([45]) Self-Determination Theory (SDT) and [21]’s ([21]) Growth Mindset (GM), provide valuable insights into autonomy, competence, relatedness, and the malleability of abilities. However, these models do not fully address the combination of goal-directed cognition and enduring resilience required for sustained re-engagement among emerging adults facing systemic inequities. Hope Theory offers a robust, evidence-based cognitive process for setting meaningful goals, identifying viable pathways, and sustaining agency ([26]; [35]; [49]; [54]). Infinite Mindset complements this by emphasizing existential flexibility, a just cause, and the capacity to navigate challenges as part of an ongoing process rather than a finite endpoint ([52]).

The integration of these two models contributes three distinct advancements. First, it unites complementary theories, one of which is grounded in psychological mechanisms of motivation and the other in adaptive, purpose-driven leadership. Second, it produces a psychologically and ethically grounded framework that supports sustained connection rather than short-term compliance or credentialing. Third, it offers evaluative guidance recognizing internal transformation and contextual adaptation as critical progress indicators.

Disconnected youth often experience a lack of hope that signals disrupted agency and diminished belief in possible outcomes rather than an absence of aspiration ([9]; [59]). Research and practice affirm the transformative power of hope, showing that targeted interventions can restore it by clarifying goals, strengthening agency, and reshaping narratives to emphasize possibility ([9]; [16]). When paired with Infinite Mindset, these interventions also develop the resilience, adaptability, and future orientation necessary to sustain motivation in the face of systemic barriers.

Simon Sinek’s Infinite Mindset offers a vital worldview for reimagining youth development. Built upon [15]’s ([15]) theory of finite and infinite games, [52] ([52]) redefines leadership as a long-term commitment to meaningful purpose rather than short-term victories. He proposes that individuals and organizations thrive when guided by the following five enduring principles:Advancing a Just Cause: A future-focused vision that lends significance to everyday action. The IH framework links individual aspirations to broader ethical considerations in its goal-setting, aligning with sociological understandings of the capacity to aspire;Building Trusting Teams: Cultivating safe, affirming spaces where people can take risks, express vulnerability, and grow. For disconnected youth, this relational safety is essential to developing hope and reflects community-level collective efficacy and social capital mechanisms;Embracing Existential Flexibility: A readiness to pivot or abandon successful strategies when a more aligned path emerges. This readiness strengthens long-term planning by encouraging intentional change and draws on the concept of capabilities as adaptive resources;Learning from Worthy Rivals: Seeing others not as threats but as mirrors for personal growth. This element reframes competition into self-reflection and supports a growth-oriented agency;Leading with Courage: Choosing principled action even when outcomes are uncertain. This component reinforces a moral backbone, allowing youth to act despite fear, and connects with reflexivity and value-aligned identity work.

These five principles are not merely abstract values. When aligned with Snyder’s Hope Theory, they become a practical guide for youth development. Hope Theory defines hope as a cognitive-motivational process consisting of the hope triad. The IH model builds on this foundation by linking goals, pathways, and agency to sociological anchors, while integrating ethical purpose and adaptive flexibility as key components of sustained motivation.

Table 1 clarifies how Sinek’s principles operate within this framework by providing concise operational definitions that translate each concept into measurable terms relevant for youth development settings. Table 2 expands this integration by mapping Hope Theory and Infinite Mindset to sociological anchors and validated measures, creating a one-to-one alignment that makes testing and implementation viable.

These leadership principles serve as the scaffolding for IH, but their strength emerges in synthesis with Hope Theory and sociological anchors. Table 2 compares Snyder’s cognitive-motivational model of hope with Sinek’s infinite mindset philosophy, mapping each element to sociological constructs and potential measures. This synthesis shows how IH weaves the hope triad, moral clarity, adaptability, and collective efficacy into a developmental strategy for young adults facing disconnection from education, employment, or training, offering both theoretical grounding and pathways for empirical validation.

## 2. Conceptual Integration

### 2.1. Empirical Grounding

A substantial body of research affirms the central role of hope in fostering positive developmental outcomes during emerging adulthood. Young people with elevated levels of hope show stronger academic performance, greater emotional regulation, and sustained persistence through adversity ([26]; [35]; [54]). These individuals express meaningful goals, identify multiple strategies for achieving them, and sustain the motivation for progress. For those experiencing disconnection, hope often diminishes, not because of an absence of aspiration, but in response to repeated systemic barriers that compromise their belief in future possibilities ([9]; [59]).

Recent empirical studies deepen this understanding and support key propositions within the IH framework. [13] ([13]) demonstrated that a strong sense of purpose enhances psychological resilience and goal-directed behavior in youth, affirming IH’s emphasis on courageous leadership and sustained agency. [35] ([35]) found that structured interventions increase future-oriented thinking and strengthen a sense of control, reinforcing the framework’s focus on adaptive pathways and value-driven motivation. The philosophical propositions of Infinite [52]’s ([52]) assertion that purposeful leadership and moral clarity drive sustainable change further support these findings ([23]).

Additional studies extend and refine these propositions. [44]’s ([44]) Critical Consciousness (CC) theory defines how young people recognize and challenge systemic inequities while developing the motivation and capacity to pursue social change. This perspective complements IH by linking cognitive awareness with purposeful action. [29] ([29]) connected critical consciousness with purpose-driven behavior, showing that when young people align their goals with a just cause, their motivation and identity clarity increase. [9] ([9]) proved that identity-affirming environments foster greater hope and persistence, reinforcing IH’s claim that personal goals gain power when connected to social purpose. [43] ([43]) demonstrated that affirming peer spaces, such as gender–sexuality alliance groups, strengthen agency and belonging. [37] ([37]) highlighted how collective hope and agency emerge through community-based processes, while [2]’s ([2]) work on the capacity to aspire links purpose to broader notions of just cause. These findings stress that trusting teams and group-level processes reinforce hope through collective efficacy and shared action.

These patterns provide conceptual scaffolds and open practical avenues for testing. The IH model encourages researchers to explore original combinations of existing constructs. Researchers can use [53]’s ([53]) State Hope Scale to assess baseline levels of goal-directed agency and pathways thinking. For constructs related to an infinite mindset, they may apply relevant tools in organizational psychology that emphasize purpose, values alignment, and ethical leadership. These tools function as interim proxies while IH-specific measures continue to emerge ([52]).

This body of work shows that hope is not an isolated emotion or a static trait. Individuals cultivate hope through coherence of identity, alignment with deeply held values, and purposeful engagement in action ([52]; [54]). The IH framework emphasizes that disconnected emerging adults do more than build skills. They reclaim belief, clarify their values, and align their direction with a sense of ethical purpose. Researchers can deepen this model by connecting hope assessments to longitudinal measures of identity development, moral leadership, and sustained purpose over time.

### 2.2. Integrated Model of Infinite Hope

The IH framework emerges from a foundation of empirical research and theoretical insight. It unifies two powerful developmental approaches: Snyder’s cognitive theory of hope and Sinek’s infinite mindset. They create a model that activates internal capacities and outward expressions of purpose-driven growth. The visual models provide a conceptual synthesis of this integration, showing how the core components of IH interconnect and develop across time. 

Figure 1 presents a three-circle Venn diagram that illustrates how the core elements of Snyder’s theory—goals, pathways, and agency—intersect with Sinek’s principles of the infinite mindset: just cause, existential flexibility, and courageous leadership ([52]; [54]). Each intersection transforms individual capacities into integrated strengths. For example, aligning goals with a just cause elevates them beyond personal ambition, imbuing them with ethical and communal significance. Purpose operates not just as a destination but as a compass, offering directional clarity grounded in values and future vision ([11]; [62]). Likewise, existential flexibility shapes adaptive pathways, empowering young people to revise strategies while staying rooted in long-term goals ([52]; [54]). Courageous leadership fortifies agency, allowing individuals to act with integrity amid fear or uncertainty ([20]; [21]).

This model shows how goals, pathways, and agency converge around an ethically sound purpose aligned with identity. Youth begin to align hope with deeper values and long-term vision. These intersections reveal more than conceptual compatibility; they generate developmental synergy. Each pairing enables hope to function as a coping mechanism and a transformative process rooted in core values and guided by vision ([52]; [54]). From a broader social perspective, goals connect to [2]’s ([2]) capacity to aspire, pathways align with [50]’s ([50]) and [39]’s ([39]) capability approach, and agency resonates with [22]’s ([22]) conception of temporal agency. Trusting teams reflect [46]’s ([46]) collective efficacy, while courageous leadership draws from [44]’s ([44]) framework of critical consciousness. These integrations strengthen the explanatory power of IH and situate it within established social theories.

Hope shifts from a passive belief to an active, lived experience. This transformation becomes especially meaningful for emerging adults who navigate instability, disconnection, or marginalization. When immersed in these integrated frameworks, young people reinterpret their past, clarify their direction, and actively shape their future with confidence and conviction ([9]; [11]; [37]).

One young adult, reflecting on the initial realization of this shift, might say, “I used to set goals without really knowing why. Now I know what matters to me, and that’s what keeps me going.” This statement shows an early stage of transformation, where new clarity of purpose grounds hope. As development progresses and self-understanding deepens, a more advanced insight may emerge. Another participant might explain, “I realized I do not just want to get by. I want to live with purpose.” This second reflection captures the internalization of purpose and values-aligned identity, demonstrating a more stable embodiment of IH.

Figure 2 presents a concentric circles model that illustrates how program features align with measurable constructs to support the development of IH. The outer ring highlights foundational skills such as goal setting, future envisioning, managing setbacks, and cultivating self-belief. These capacities correspond to identity salience and early markers of self-regulation, which intake assessments can capture as baseline indicators ([55]). The model shows that these skills provide the momentum for sustained engagement and signal areas where support programs should first concentrate.

The middle ring demonstrates foundational capacities deepen when youth connect their goals to broader meaning and purpose. This layer reflects constructs such as purpose scales ([11]) and measures of adaptive flexibility, which are essential for tracing developmental progress. By linking program features like mentorship, reflective exercises, and collective problem-solving to these constructs, evaluators can measure immediate outcomes and longer-term alignment with IH. For example, trusted peer teams and community-building activities align with collective efficacy ([46]), strengthening pathways that sustain agency and hope.

At the center, IH represents a durable internal state defined by values-aligned identity and courageous leadership. Instruments such as critical consciousness scales ([44]) and collective efficacy measures ([46]) help document how participants translate internal growth into relational and social action. By incorporating these measures into intake and outcome assessments, programs ensure that evaluation reflects the full arc of development from initial skills through purposeful agency. Expanded descriptions of Figure 1 and Figure 2 appear in Appendix A.

### 2.3. Implications for Practice and Program Design

The IH framework urges youth practitioners to address the dynamic interaction between personal development and contextual supports ([40]; [52]; [54]). Authentic engagement begins with meaning rather than control or compliance. Programs must move beyond transactional models to cultivate belief, reinforce agency, and center purpose. These conditions restore young persons’ relationships with their futures by aligning youth strengths with tangible support and ecological assets.

Evidence supports this integrated approach. [14] ([14]), [35] ([35]), [47] ([47]), and [54] ([54]) demonstrate that emotionally present and structurally supportive adults, whether or not they are formal mentors, foster clarity and momentum in youth development. [43] ([43]) found that affirming peer environments, particularly those grounded in shared identities, strengthen belonging and agency. [20] ([20]) and [45] ([45]) show that autonomy and connectedness nurture internal motivation, making sustained purposeful action more likely. Hope grows through relationships; it thrives when others see, affirm, and support young people.

Programs informed by IH must center cultural identity, validate lived experience, and integrate context-specific support such as housing assistance, mentorship, transportation access, safe spaces, and mindset-shifting and goal-setting strategies ([52]; [54]). Youth-led organizing, supportive peer spaces ([43]), trauma-attuned learning environments ([29]), and critical consciousness-building strategies ([44]) help youth attach meaning to their narratives and act on their goals.

This model calls for redefining success ([52]; [54]). Attendance, credential attainment, and completion rates offer limited insight. IH encourages measuring deeper transformation by recognizing agency, goal clarity, value alignment, and purposeful action. [9] ([9]) emphasize that re-engagement is most effective when grounded in purpose and psychological empowerment. [62] ([62]) confirm that respectful, affirming environments promote constructive feedback uptake and sustained growth.

Bringing IH into practice requires translating each component into actionable strategies ([52]; [54]). The following table links program features to measurable constructs, clarifying how intake and outcome assessments can align with IH principles.

Table 3 presents programmatic strategies that operationalize the IH framework by directly connecting constructs from Hope Theory and Infinite Mindset to practical applications in program design and facilitation ([52]; [54]). Each pairing of constructs offers clear, actionable strategies that practitioners can embed into their engagement models to foster sustained connection, reflection, and future orientation among youth.

Programs to re-engage disconnected youth must prepare staff to function as more than instructional deliverers. Staff should act as developmental guides, modeling purpose, reinforcing agency, and embodying the integrity they hope to inspire ([4]; [10]; [23]; [52]; [54]). Organizations strengthen their ability to activate IH by promoting a reflective, purpose-aligned culture and providing structured, experience-based professional development rooted in youth identity, resilience, and the integrated principles of the framework. Staff become co-constructors of growth, translating theory into relational interactions that help youth reclaim their future with clarity and direction (see Appendix B for recommended training components).

Professional learning must be ongoing, beginning with robust onboarding and continuing through reflective practice. Training in trauma-responsive communication, cultural humility, motivational interviewing, purpose-based advising, and critical consciousness development enhances practitioner impact ([20]; [29]; [33]; [35]; [44]). When adults model adaptability, long-range thinking, and courage, they reinforce the qualities they aim to cultivate in young people.

The success of IH rests on staff consistently embodying its principles. Consistent relational support and affirmation catalyze psychological transformation ([16]; [20]; [28]; [29]). Core competencies should include clarification of values, exploration of identity, facilitation of purpose, and narrative listening. Practitioners should integrate structured journaling or storytelling as a reflective tool, enabling emerging adults to critically engage with their past, articulate current intentions, and imagine future possibilities ([11]; [19]; [31]; [32]; [36]). Reflection builds metacognitive awareness and narrative coherence, essential to resilience and hope ([11]; [54]). Safe, supportive environments and thoughtful prompts help transform past disruptions into purposeful engagement (see Appendix C). Additional applied strategies for embedding IH principles into instructional, relational, and reflective practices are provided in Appendix D.

## 3. Method

### 3.1. Conceptual Framing and Review Methodology

The IH framework builds on empirical research, established theories, and applied practices across psychology, education, youth development, leadership studies, and sociology. We integrate these models to form a unified approach that shows how disconnected emerging adults can re-engage with purpose, direction, and possibility. Instead of generating new data, the framework reinterprets existing scholarship to propose an interdisciplinary and practice-oriented approach grounded in lived experience.

The review used narrative synthesis to collect and interpret insights across disciplines (see Table 4). The researcher searched ProQuest Central, EBSCOhost Academic Search Premier, PsycINFO, ERIC, Scopus, Web of Science, Google Scholar, and key sociology journals. They expanded coverage through forward and backward citation tracing and manual reviews of leading journals. The review concentrated on English-language, peer-reviewed publications that addressed emerging adults aged 18–25 who experienced disconnection from education or employment.

The keyword strategy included terms related to hope theory, agency, identity development, purpose, infinite mindset, narrative identity, capabilities, collective efficacy, and critical consciousness. This comprehensive search identified 569 records. After screening, reviewers confirmed no duplicates and excluded twenty-three sources that fell outside the scope or lacked sufficient publication details, leaving a final set of 546 peer-reviewed studies. Influential contributions include [10] ([10]), [11] ([11]), [18] ([18]), [23] ([23]), [29] ([29]), [40] ([40]), [44] ([44]), [46] ([46]), [52] ([52]), [54] ([54]), and [55] ([55]). The researcher extracted and organized measures according to the categories in Appendix E (i.e., agency, capabilities, identity, and collective efficacy) to ensure conceptual clarity and practical consistency.

This study reviewed the gray literature, such as practitioner reflections and implementation models, to provide context, but intentionally excluded it from the final synthesis to preserve rigor and rely solely on peer-reviewed scholarship.

Figure 3 displays the PRISMA-Lite flow diagram, outlining the number of records identified, exclusions applied, final studies assessed, and coding categories employed.

The synthesis shows that sustainable re-engagement requires more than surface-level behavioral change. Identity alignment, clarity of values, and purposeful connection drive the psychological transformation that supports long-term re-engagement. The resulting conceptual model strengthens developmental theory while also highlighting the applied utility of IH.

### 3.2. Positioning Infinite Hope Among Established Motivational Theories

IH stands within the broader tradition of motivational theory and draws on Relational Developmental Systems Metatheory (RDS) ([40]) to emphasize ecological development. [45]’s ([45]) Self-Determination Theory (SDT) and [41]’s ([41]) Control-Value Theory (CVT) provide points of overlap, contrast, and added value. SDT highlights autonomy, competence, and relatedness as universal needs that fuel motivation. CVT explains how perceived control over tasks and their value shape emotional experience and self-regulated behavior ([41]). Relational approaches to hope, described by [14] ([14]) and [47] ([47]), show how supportive relationships nurture agency, pathways thinking, and purpose.

IH combines these insights while embedding Critical Consciousness (CC) to address marginalized youth development’s cognitive, motivational, and behavioral dimensions ([44]). Agency parallels SDT’s autonomy and CVT’s control appraisals ([23]; [35]; [54]). Pathways thinking reflects competence and adaptive control strategies ([41]). Goal setting tied to a just cause extends intrinsic motivation and strengthens the value dimension central to CVT ([42]). Existential flexibility highlights moral adaptability, a feature rarely emphasized in SDT or CVT, yet essential for identity reconstruction and resilience ([18]). CC sharpens these constructs by equipping youth to examine systemic inequities and act for change. Sociological anchors root IH in traditions of collective efficacy ([46]), identity theory ([55]), and critical reflection/action ([44]).

Table 5 compares IH constructs with SDT, CVT, RDS, and CC while mapping each construct to its sociological anchor and measures. This alignment demonstrates how IH bridges motivational theory with approaches grounded in identity, values, social context, and structural awareness. It also addresses critiques of ill-defined constructs and overlap by showing clear anchors and validated tools.

Integrating cognitive motivation, emotional regulation, ethical orientation, relational context, critical social awareness, and sociological grounding gives IH its distinctive strength ([14]; [40]; [44]; [46]; [55]). Hope-based constructs ([35]; [54]) and leadership principles from Infinite Mindset ([52]) reinforce these dimensions, creating a framework that unites psychological, sociological, and adaptive leadership capacities. While SDT emphasizes psychological needs and CVT highlights achievement emotions, IH links personal striving with purpose-driven leadership, identity clarity, moral vision, and community capacity.

The framework defines hope as a goal-directed trait and a socially constructed, ethically grounded practice. Individuals cultivate it through adaptive strategy, community support, and principled courage. This alignment affirms IH’s conceptual coherence and expands its usefulness. It builds on psychological and sociological theory while offering a values-based model of agency and purpose designed for young adults navigating uncertainty, transition, and systemic barriers.

### 3.3. Literature Integration and Theoretical Anchoring

Hope Theory anchors IH in psychological theory. [53] ([53]) and later [54] ([54]) defined hope as a cognitive–motivational structure composed of the hope triad. These elements enable individuals to envision meaningful futures and pursue them intentionally. Prolonged adversity often erodes hope among disconnected emerging adults. However, research demonstrates that targeted interventions restoring goal clarity, enhancing adaptive thinking, and fostering self-efficacy can rebuild hope and renew purposeful engagement ([26]; [35]; [52]; [59]).

[52]’s ([52]) Infinite Mindset, inspired by [15]’s ([15]) theory of finite and infinite games, adds philosophical and ethical depth to this model. Sinek identified five Infinite Mindset principles, each reinforcing a dimension of Hope Theory ([54]). Existential flexibility, for example, broadens pathways by validating moral redirection. Leading with courage strengthens agency through value-aligned action. Advancing a just cause elevates goals into a collective, ethical endeavor.

Although Infinite Mindset lacks formal validation in developmental psychology, its philosophical clarity and alignment with Hope Theory and recent studies on values-based motivation underscore its relevance ([35]; [52]; [54]). Its core principles resonate with developmental priorities such as meaning-making, moral identity, and resilience in uncertainty. Researchers should prioritize psychometric development, mixed-methods testing, longitudinal analyses of goal persistence, emotional adaptability, and narrative repair. These methods offer the empirical tools necessary to test, refine, and validate IH’s theoretical claims across diverse developmental contexts.

### 3.4. Scope and Relevance of Literature

The literature reviewed spans developmental psychology, educational leadership, civic engagement, and youth development. This interdisciplinary foundation reflects the complexity of disconnection and the diverse strategies required for meaningful and sustainable re-engagement ([54]). This review integrates foundational and emerging contributions to show how hope, mindset, identity, critical consciousness, and relational systems operate within transformative interventions. It also highlights debates, critiques, and measurement implications that inform IH.

Table 6 presents a thematic summary of key concepts, their propositions, and areas of scholarly divergence. It contrasts perspectives on collective efficacy, social capital, and community cultural wealth while noting implications for measurement. By organizing the literature into theoretical domains, the table clarifies the framework’s grounding and emphasizes unresolved tensions that shape future research and practice ([52]; [54]). This synthesis affirms IH’s role in advancing integrated, forward-looking approaches to youth development.

These perspectives show that hope emerges from personal willpower and relational systems, cultural strengths, and structural contexts ([23]; [38]). Collective efficacy, social capital, and community cultural wealth remain contested concepts, yet each offers insight into how emerging adults navigate re-engagement. This tension highlights the need to triangulate measures across psychological and sociological domains for IH. Doing so ensures that assessments capture both internal change and external supports that sustain purposeful growth.

Building on this conceptual foundation, this study applies an interdisciplinary literature review to explore how hope, agency, purpose, relational systems, and critical consciousness shape emerging adulthood ([35]; [40]; [44]; [52]; [54]). Rather than collecting new data, this study synthesizes existing theoretical and empirical findings. Guided by IH, it examines how internal motivation and external context influence future orientation. The following section outlines the research methods used to structure this integrative analysis.

## 4. Discussion

### 4.1. Rethinking Disconnection as Developmental Interruption

The IH framework redefines disconnection as more than a lapse in academic or professional engagement; it disrupts developmental continuity and identity-building. Rather than treating disconnection as an individual shortcoming, IH situates it within broader social, economic, and institutional structures that restrict opportunity. This lens clarifies why strategies that rely only on behavioral compliance or credential attainment often fail; they ignore structural inequities, cultural displacement, and resource scarcity that weaken agency, purpose, and motivation ([8]; [9]; [12]).

Disconnection interrupts the continuity of a young person’s life story, diminishing belief in achievable futures. These breaks erode the core capacities emphasized in Hope Theory and Infinite Mindset—agency, goal clarity, and future orientation—needed for resilient development ([52]; [54]). Reconnection succeeds when environments affirm lived experience, encourage critical reflection, and restore coherence in identity ([19]; [31]; [38]).

The framework addresses these realities by integrating systemic barriers and limited opportunities into its analysis. Conditions such as unstable housing, inequitable schooling, and reduced access to supportive networks directly contribute to youth disengagement ([10]; [23]). Programs provide practical support and anchor psychological growth opportunities that sustain participation.

Relational Developmental Systems Metatheory reinforces this ecological view by showing how individual development constantly interacts with contextual resources ([40]). IH extends this idea by illustrating how synergy between youth strengths and ecological assets restores developmental momentum. When relational scaffolds and structural supports align, they create conditions where agency, purpose, and motivation can take root. This alignment also strengthens a shared belief in community capacity, enhancing individual confidence and collective progress.

### 4.2. Reclaiming Motivation and Forward Momentum Through Hope

Hope Theory drives the IH model by explaining how young people move from disorientation to goal-directed action. Instead of measuring progress only by external benchmarks, Hope Theory emphasizes internal mechanisms: the ability to set goals, design pathways, and sustain belief in personal capacity to reach them ([54]). When these mechanisms align, hope grows and self-concept evolves.

Within the IH framework, programs cultivate these dynamics by offering structured yet adaptable tools such as narrative reframing, purpose mapping, and scaffolded goal setting. These practices gain transformative impact in environments that encourage emotional regulation, peer validation, and reflective growth ([14]; [32]; [35]). Relational perspectives highlight that hope develops through individual cognition and supportive social connections that validate agency and purpose ([47]). Such practices sustain agency by reinforcing that actions carry meaning and open future possibilities.

IH positions hope as a renewable developmental resource rather than a fleeting emotional state. Practitioners can teach, model, and strengthen hope through consistent, growth-oriented relationships. When young people reinterpret past interruptions as sources of learning rather than failure, they shift toward possibility and direction ([11]; [16]). Hope becomes a bridge that links disrupted experiences to renewed purpose, equipping young people with the mindset and networks to build a coherent and empowered future. This process also deepens their sense of agency across time, supporting what can be called forward-looking persistence, or the ability to envision and sustain long-term pathways despite obstacles.

### 4.3. Navigating Youth Agency in the Age of AI and Digital Systems

Digital environments increasingly influence how emerging adults shape identity, exercise agency, and pursue future goals. IH addresses these realities by examining how digital tools intersect with broader structural barriers restricting opportunity. Artificial intelligence (AI) and algorithmic systems expand or constrain autonomy, purpose, and decision-making. At the same time, inequities in access to supportive resources intensify disengagement. Situating personal growth within these external conditions, IH shows that agency grows most fully when technological and systemic factors align ([40]).

[51] ([51]) shows that AI-powered learning environments enhance motivation and self-efficacy by focusing on user control, relevance, and transparent feedback. Platforms that allow youth to track progress, connect tasks to goals, and receive timely input strengthen purpose and confidence. Equally important, programs that provide mentoring, transportation, or safe communal spaces offset systemic disadvantages and create conditions where agency and hope can flourish ([14]; [47]).

IH emphasizes that digital engagement must involve more than providing access to devices. Programs should help youth develop digital discernment and the ability to evaluate how data-driven systems filter opportunities, influence identity, and reinforce inequities. This discernment and structural awareness equips youth to navigate both digital and social environments with critical consciousness ([44]). [58] ([58]) reinforce this view by framing digital participation as an ongoing meaning-making process through iterative learning.

Programs that confront systemic barriers alongside digital contexts expand youth agency. Reflective practices such as journaling, storytelling, and goal mapping help participants process experiences, clarify aspirations, and monitor progress across physical and digital domains. Combined with concrete resources and relational supports, these strategies affirm dignity, strengthen direction, and build resilience ([19]; [31]). IH shows that youth agency thrives when structural resources and purposeful relationships work together to support development.

### 4.4. Infinite Mindset: Shaping Identity Through Enduring Purpose

IH offers a strategic framework for sustaining identity growth and purposeful engagement amid uncertainty. While Snyder’s Hope Theory illustrates how youth progress through agency, pathways, and goals, the IH Mindset clarifies why they persist. It guides young people toward futures anchored in ethical clarity and enduring meaning rather than short-term outcomes ([54]; [52]).

Sinek’s five principles address critical developmental needs. A just cause encourages youth to define goals beyond survival and affirm contribution ([11]; [62]). Trusting teams provide psychological safety and belonging, which are vital for youth navigating instability ([43]). Adaptive purpose cultivates flexibility, allowing youth to revise strategies while sustaining values ([17]). Learning from rivals fosters humility and resilience through comparison and reflection ([37]). Leading with courage strengthens moral identity and nurtures confidence to act even amid social and emotional risk ([44]; [48]).

Evidence from youth-led contexts illustrates the transformative impact of these principles. [37] ([37]) describe how activist collectives foster ethical self-authorship, enabling young people to align action with belief and reframe identity through civic purpose. Similarly, [43] ([43]) show how identity-affirming spaces elevate belonging, well-being, and hope. These contexts function as ecosystems of transformation that help emerging adults reconstruct aspirations, internalize agency, and root identity in contribution ([23]; [43]).

Practitioners who cultivate an IH Mindset help young people sustain hope by grounding it in narrative coherence and ethical direction. Rather than replacing Snyder’s framework of agency, pathways, and goals, the IH Mindset extends it into a broader ethos of intentional living. Infinite thinking redefines resilience as proactive growth rooted in long-range purpose, adaptive strategies, and value alignment ([17]; [34]; [52]). Setbacks become catalysts for learning and growth rather than markers of inadequacy ([9]; [62]).

Adopting an IH Mindset enables disconnected youth to build a foundation for sustained meaning-making. This approach anchors goals in values, strengthens intentional decision-making, and fosters narrative healing across past, present, and envisioned futures ([11]; [62]). As youth embrace clarity, conviction, and courage, they move from reactive persistence to purpose-driven transformation, redefining identity with agency and sustaining hope with integrity.

### 4.5. From Framework to Practice: Integrating Hope and Mindset for Transformative Growth

IH offers a forward-looking conceptual framework rooted in positive youth development (PYD), self-determination theory (SDT), control-value theory (CVT), and hope theory. Research shows that young people thrive when they build psychological strengths such as agency, self-determination, and future orientation ([11]; [35]; [54]). IH extends this knowledge by focusing on the unique challenges of older emerging adults disengaging from traditional pathways. It emphasizes critical reflection, narrative identity, adaptive purpose, and critical consciousness, aligning with the complex realities these young people face when navigating structural barriers and disrupted transitions ([44]). Like any evolving model, IH requires testing, refinement, and validation. Its strength also rests in sociological foundations that link individual agency to structural conditions and community trust ([46]; [55]; [44]). Current support draws largely from qualitative studies and pilot programs that provide deep insights but remain limited in generalizability ([8]; [9]).

Future researchers should expand the evidence base by addressing measurement validity, cultural responsiveness, and practitioner training. Standardized yet adaptable tools can ensure fidelity across diverse settings, while culturally responsive strategies can increase scalability. Programs that lack these supports risk inconsistent application or a shallow interpretation of IH’s developmental depth.

Stronger grounding requires longitudinal, cross-cultural, and participatory research that examines how systemic conditions shape outcomes ([4]; [11]; [40]; [56]). Hope and purpose may function differently depending on youth access to networks, ecological resources, and community trust ([12]; [14]; [47]). Effective implementation tailors IH to reflect local assets and diverse youth identities ([24]).

Evaluation should reach beyond external benchmarks such as employment or credential attainment to include developmental shifts like identity coherence, cognitive flexibility, moral clarity, and future orientation. Appendix E provides details of validated measures, including the Revised Youth Purpose Survey ([11]), Self-Transcendent Purpose for Learning Scale ([62]), Hope Scale subscales ([54]), General Self-Efficacy Scale ([48]), Cognitive Flexibility Inventory ([17]), and Dimensions of Identity Development Scale ([34]). Practitioners and researchers can use Appendix E as a flexible guide, selecting and triangulating measures that fit their context. When paired with narrative interviews, reflective journals, and participatory inquiry, these tools capture the layered architecture of hope.

IH also benefits from deeper alignment with broader developmental theory. While it builds on Snyder’s hope theory and Carse’s concept of infinite games ([15]; [52]), it connects with Arnett’s emerging adulthood framework, RDS ([40]), and identity development models that highlight culture and context ([5]). Constructs such as existential flexibility and clarity of purpose need sharper operationalization to ensure reliable use in research and practice.

Qualitative and mixed-methods approaches offer effective pathways for testing and adapting IH. Narrative interviews, reflective journaling, and participatory inquiry elevate youth voices and honor the internal transformations often preceding external progress ([8]; [27]). Longitudinal studies can show how changes in hope, mindset, and critical reflection influence outcomes like education, employment, civic participation, and well-being ([33]; [38]). These designs reveal how agency, identity, and purpose develop and how social contexts sustain or hinder growth.

As IH evolves, practitioners must recognize cultural variation in how young people define and express hope ([14]; [23]; [29]). Implementation in under-resourced settings depends on strong relationships, mentorship, transportation, safe spaces, ethical presence, and institutional investment. Programs lacking these supports risk misapplying IH as a compliance tool rather than a pathway to empowerment ([24]).

These opportunities highlight IH’s potential and the importance of continued research, collaborative testing, and youth co-design. Each study and lived experience adds depth to the model, enriching it as a tool for re-engaging disconnected youth. As [4] ([4]) emphasized, emerging adulthood is a period of possibility. IH expands that possibility by equipping young people to imagine, construct, and pursue futures grounded in agency, purpose, and integrity. Appendix F provides a research agenda with testable hypotheses that align with the framework’s dimensions.

## 5. Conclusions

IH offers more than a synthesis of theory. It presents a vision for how emerging adults can strengthen agency, clarify purpose, and rebuild resilience. When supported by the right opportunities and relationships, IH reframes disconnection as an inflection point that can ignite growth, renewal, and re-engagement.

Programs affirm identity and create coherence between goals and purpose. They create belonging and direction, moving beyond behavioral compliance toward transformational, youth-centered approaches highlighting potential and sustaining engagement.

IH calls on educators, practitioners, researchers, and policymakers to extend evaluation beyond employment or credential attainment to include purpose, clarity of vision, and motivation. Programs that affirm identity, nurture reflection, and build agency prepare young people to confidently shape their stories. Youth thrive when they recognize their value, trust their voice, and act with intention.

This shift connects directly to national initiatives prioritizing developmental growth and equity. Networks such as the Reconnecting Youth Campaign ([57]), the Aspen Institute’s Opportunity Youth Forum ([6]), Forum for Youth Investment ([25]), YouthBuild ([64]), and Jobs for the Future ([30]) provide models that demonstrate consistency with IH principles. These organizations show how policy and practice can converge to build developmental ecosystems where young people can aspire, connect, and lead.

IH also advances a testable pathway grounded in social and developmental theory: goals as capacity to aspire, pathways as capabilities, agency as temporal orientation, trust and cooperation as shared efficacy, and courage as critical consciousness. These foundations illustrate how cognition, emotion, relationships, and context intersect to support agency and resilience.

In centering emerging adults on strengths and potential, IH affirms lived experience and the capacity for change. It offers a developmental and ecological orientation that reconnects youth with goals, redefines identity, and opens pathways toward purposeful futures.

Policy recommendations flow directly from this synthesis. National models such as the Reconnecting Youth Campaign, Aspen’s Opportunity Youth Forum, Forum for Youth Investment, YouthBuild, and Jobs for the Future illustrate how coordinated policy and practice can create supportive ecosystems. Each of these initiatives shows a distinct approach: national advocacy for reconnecting youth, community-driven forums for shared learning, investment in cross-sector youth development, workforce training through YouthBuild, and educational–employment bridges through Jobs for the Future (JFF). States and local jurisdictions will inevitably adapt these strategies to fit their demographics, resources, and priorities. Therefore, practitioners and policymakers should interpret IH principles as adaptable templates, assessing how best to integrate them into their communities and systems. For example, state workforce boards may incorporate IH by embedding purpose-driven goal setting into career pathway initiatives, while education compacts could adopt IH principles by ensuring postsecondary transitions reflect youth voice and agency. Local governments may apply IH through cross-sector collaboratives that connect housing, education, and employment supports in ways that emphasize identity, resilience, and purpose.

Future research should complement these policy strategies by piloting and testing IH in diverse community contexts, examining how hope, an infinite mindset, and contextual resources interact across time, identity groups, and developmental environments to refine practice and inform scalable strategies. IH calls for collective action that unites research, practice, and policy in serving emerging adults. The work ahead demands courage to rethink entrenched systems, flexibility to adapt to local needs, and commitment to build environments where every young adult can cultivate purpose and pursue possibility. Readers should view these recommendations not as prescriptions but as opportunities—tools to shape strategies that reflect their communities’ unique strengths and challenges while advancing the shared goal of resilient, thriving emerging adults.

## Figures and Tables

**Figure 1 behavsci-15-01205-f001:**
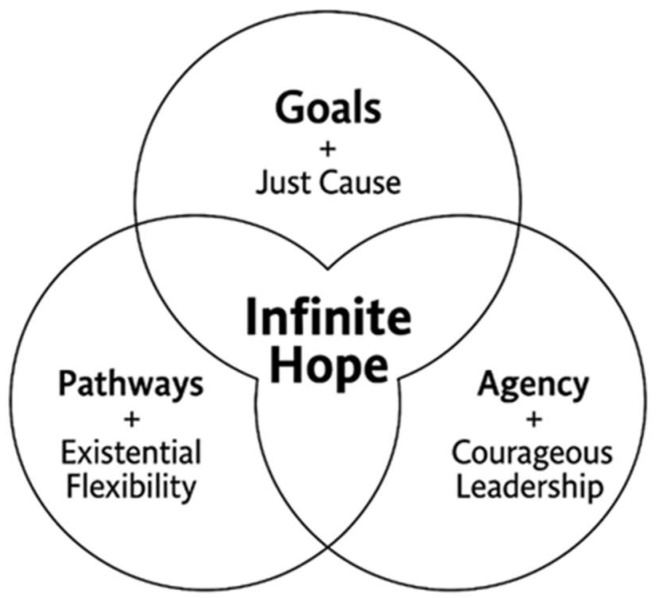
The Infinite Hope Framework (Venn Diagram Model). Adapted from [52] ([52]) and [54] ([54]).

**Figure 2 behavsci-15-01205-f002:**
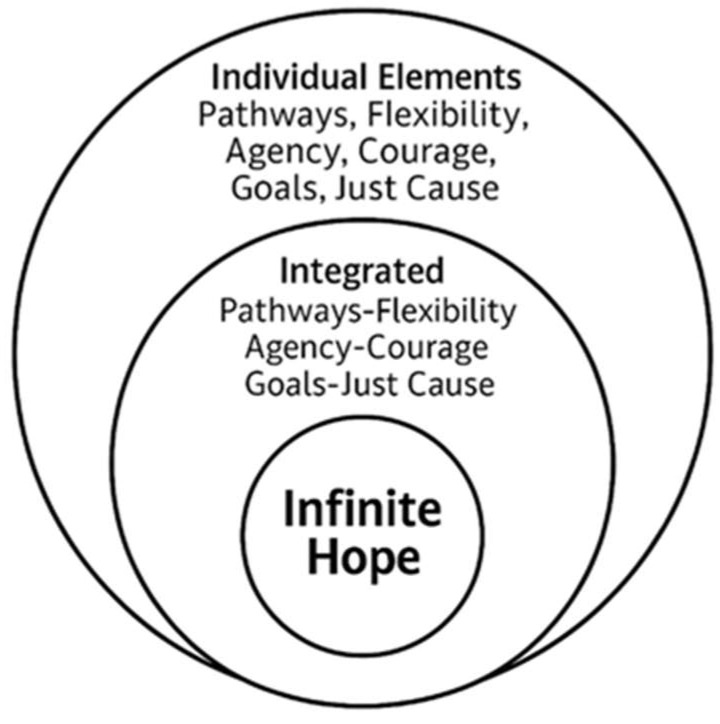
Developmental Growth Through Infinite Hope (Concentric Circles Model). Adapted from [52] ([52]) and [54] ([54]).

**Figure 3 behavsci-15-01205-f003:**
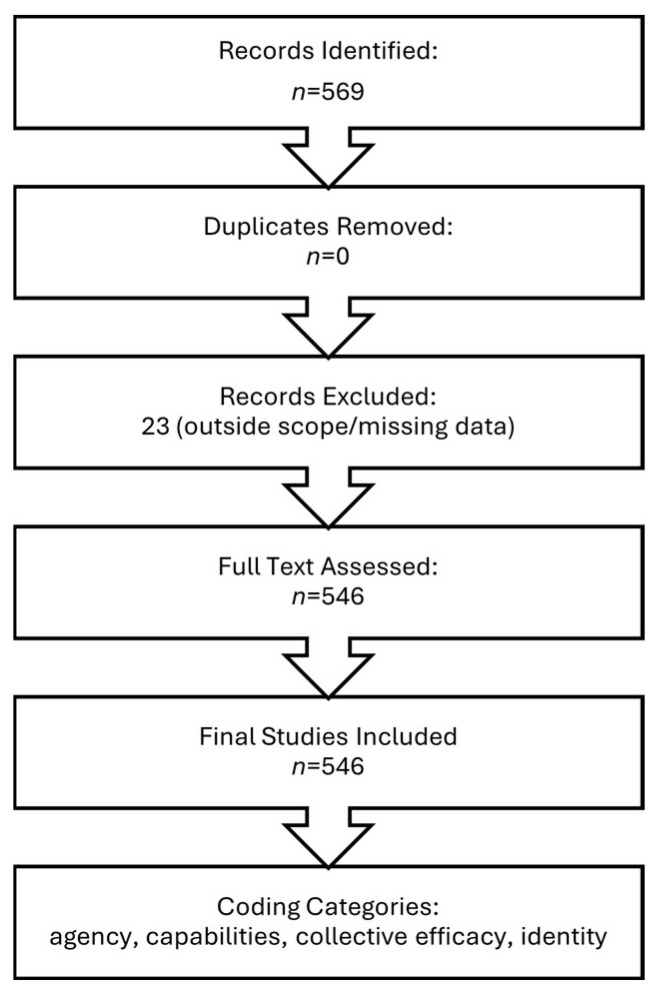
PRISMA-Lite Flow Diagram.

**Table 1 behavsci-15-01205-t001:** Operational Definitions of Sinek’s Infinite Mindset Principles within the Infinite Hope Framework.

Construct	Operational Definition	Sociological Anchor + Measure
Advancing a Just Cause	A personally meaningful, future-oriented vision that benefits others and provides ethical direction.	[2] ([2]): Capacity to Aspire—purpose as culturally scaffolded; measurable with Revised Youth Purpose Survey ([11]).
Building Trusting Teams	Creation of psychologically safe environments where individuals feel supported and respected.	[46] ([46]): Collective efficacy—trust and willingness to intervene; validated 10-item scale.
Embracing Existential Flexibility	Capacity to pivot from even successful strategies toward more aligned, meaningful goals.	[50] ([50]); [39] ([39]): Capabilities approach—adaptability as expansion of substantive freedoms; proxies via Cognitive Flexibility Inventory ([17]).
Learning from Worthy Rivals	Viewing others as sources of insight and growth, rather than competition.	[22] ([22]): Agency as projective/practical–evaluative; can be proxied with general self-efficacy and social comparison measures.
Leading with Courage	Acting in alignment with values despite risk or uncertainty.	[3] ([3]); [44] ([44]): Reflexivity and Critical Consciousness—measured with CCS-S (Short Critical Consciousness Scale).

**Table 2 behavsci-15-01205-t002:** Conceptual Comparison of Snyder’s Hope Theory, Sinek’s Infinite Mindset, and the Infinite Hope Framework.

Dimension	Snyder’s Hope Theory	Sinek’s Infinite Mindset	IH Framework	Sociological Anchor + Measure
Core Focus	Goals, pathways, agency	Purpose-driven leadership, long-term vision	Integration of motivational goal pursuit with an ethical worldview	[2] ([2])—goals as the capacity to aspire.
Agency	Belief in one’s capacity to initiate and sustain action	Courageous action aligned with values	Agency fueled by ethical conviction and flexibility	[22] ([22])—temporal agency; [3] ([3])—reflexivity.
Pathway Thinking	Identifying multiple strategies to reach goals	Pivoting when conditions change	Adaptive flexibility grounded in values	[50] ([50]); [39] ([39])—capabilities and freedom to revise strategies.
Purpose Alignment	Implied in goal pursuit	Central (Just Cause)—enduring benefit beyond self	Personal aspirations anchored in ethical vision	[11] ([11]); [2] ([2])—validated purpose measures.
Adaptability/Flexibility	Persistence emphasized	Existential flexibility central	Flexibility as moral, strategic, and values-driven	[17] ([17])—Cognitive Flexibility Inventory.
Trusting Teams	Not emphasized	Psychological safety, shared risk	Relational spaces cultivating belonging and hope	[46] ([46])—collective efficacy scale.
Courage	Motivation to sustain pathways	Principled action despite risk	Courage aligned with identity and values	[44] ([44])—Critical Consciousness (CCS-S).
Identity Alignment	Not explicit	Implied through just cause	Explicit: coherence between identity and purposeful action	[55] ([55])—identity salience/verification.

**Table 3 behavsci-15-01205-t003:** Program Features and Measurable Constructs.

Program Feature	Measurable Construct/Scale
Purpose workshops and values-based goals	Purpose Scales ([11])
Coaching on strategy revision/adaptability	Existential Flexibility/Capabilities Measures ([50]; [39])
Leadership under uncertainty	Critical Consciousness Scale (CCS-S) ([44])
Capstone projects on identity/purpose	Identity Salience and Coherence ([55])
Long-term coaching and alumni engagement	Collective Efficacy ([46])

**Table 4 behavsci-15-01205-t004:** Narrative Review Search Strategy (PRISMA-Lite format).

PRISMA-Lite Element	Operational Detail
Review Objective	Integrate Snyder’s Hope Theory with Sinek’s Infinite Mindset to build the IH conceptual framework for re-engaging disconnected emerging adults.
Timeframe	January 2000–April 2025 (inclusive)
Databases	ProQuest Central; EBSCOhost Academic Search Premier; PsycINFO; ERIC; Scopus; Web of Science; Google Scholar; targeted sociology journals
Supplementary Search Methods	Forward and backward citation tracing; manual searches of top-tier journals in psychology, education, youth development, sociology, social work, and leadership studies
Search Strings (Boolean)	(“Hope Theory” OR “agency” OR “goal setting” OR purpose OR “meaning making” OR “infinite mindset” OR “identity development” OR “disconnected youth” OR “emerging adulthood” OR “institutional disconnection” OR “collective hope” OR “narrative identity” OR “critical consciousness” OR “relational developmental systems” OR Overton) AND (“youth” OR “emerging adults”) AND (disconnect* OR marginal* OR exclusion OR “systems navigation”)
Inclusion/Exclusion Criteria	Included: English-language, peer-reviewed studies on emerging adults (18–25) disconnected from education, employment, or training. Excluded: studies outside the age range, non-English, opinion-only, or lacking empirical/theoretical grounding.
Records Identified	569
Records Excluded/Removed	Duplicates: 0; Outside scope/missing publication data: 23
Full Texts Assessed for Eligibility	546
Final Studies Included	546 peer-reviewed sources. Core contributions: [54] ([54]); [52] ([52]); [11] ([11]); [10] ([10]); [23] ([23]); [44] ([44]); [46] ([46]); [55] ([55]).
Coding Categories Applied	Agency, capabilities, collective efficacy, identity
Measures Extracted	Validated scales and constructs aligned with Appendix E

(Note: This PRISMA-Lite table summarizes the narrative-review search strategy, listing the databases consulted, key search terms, coding categories, screening totals, and the final number of sources included.)

**Table 5 behavsci-15-01205-t005:** Comparative Alignment of Infinite Hope Constructs with SDT, CVT, RDS, CC and Sociological Anchors.

IH Construct	SDT Analogue	CVT Analogue	RDS Analogue	Critical Consciousness Analogue	Sociological Anchor and Measures
Agency	Autonomy ([45])	Control ([41])	Person ↔ context coaction emphasizing agency	Reflection–Action cycle for sociopolitical engagement	Identity Theory ([55]); Hope Scale—Agency Subscale ([54])
Pathways Thinking	Competence ([45])	Control & Value appraisals ([41])	Developmental plasticity and adaptive pathways	Critical reflection to identify systemic barriers	Adaptive Development (RDS plasticity); Hope Scale—Pathways Subscale ([54])
Goal + Just Cause	Intrinsic Motivation ([45])	Value appraisals ([41])	Relational purpose shaped by cultural context	Motivation linked to collective liberation and justice	Purpose Scales ([11]; [62])
Existential Flexibility	Autonomy-supportive Adaptability	Value reappraisal and emotional regulation	Person–environment adaptability and resilience	Critical action to revise strategies in inequitable systems	Cognitive Flexibility Inventory ([17])
Learning from Worthy Rivals	Relatedness through shared growth	Control through vicarious competence	Relational co-development across peers	Dialogic awareness raising within communities	Collective Socialization Models; Peer Development Measures
Leading with Courage	Integrated regulation under self-endorsed values	Value-driven challenge perception	Agency embedded in social structures	Action to resist oppression and advance equity	Collective Efficacy Scale ([46]); Authentic Leadership Questionnaire ([61])

**Table 6 behavsci-15-01205-t006:** Thematic Literature Summary of Key Constructs and Conflicting Findings.

Theoretical Model	Key Authors	Core Proposition	Contrasts or Conflicting Views	Measurement Implications for IH
Hope Theory	[54] ([54]); [16] ([16])	Hope involves cognitive motivation built on goals, pathways, and agency.	Scholars debate whether hope is a stable trait or a teachable, situational strength.	Use Hope Scale subscales to capture trait and state dimensions of hope.
Relational Developmental Systems Metatheory	[40] ([40])	Development emerges from a dynamic, reciprocal interaction of individuals and contexts.	Some argue it is too broad to serve as a predictive model without sharper operationalization.	Apply mixed-methods designs to link context-specific measures with developmental outcomes.
Critical Consciousness	[44] ([44])	Awareness of inequities and motivation to act fosters empowerment and agency.	Critics question scalability across cultural contexts and program types.	Use CCS-S (Short Critical Consciousness Scale) alongside narrative inquiry.
Self-Determination Theory	[45] ([45]); [35] ([35])	Autonomy, competence, and relatedness foster internal motivation and well-being.	Motivation may falter under structural barriers or socioeconomic pressure.	Combine SDT scales with contextual measures (e.g., socioeconomic stress indices).
Control-Value Theory	[41] ([41]); [41] ([41]); [51] ([51])	Control and perceived value shape achievement emotions.	Appraisals differ across cultures and digital learning environments.	Use CVT emotion scales and extend with digital engagement measures.
Growth Mindset	[21] ([21])	Belief in the ability to improve through effort influences motivation and learning.	Critics argue that a growth mindset oversimplifies structural barriers to success.	Pair mindset measures with structural equity assessments.
Collective Efficacy	[46] ([46])	Shared norms and trust strengthen community-level outcomes.	Advocates stress its importance for re-engagement; critics warn it may overstate communal responsibility.	Employ collective efficacy scales while triangulating with measures of access to resources.
Social Capital vs. Community Cultural Wealth	[12] ([12]); [63] ([63])	Networks and cultural wealth expand opportunities and affirm resilience.	Social capital emphasizes mainstream access; community cultural wealth centers marginalized strengths and assets.	Incorporate Yosso’s CCW framework to balance the measurement of deficits and assets.

## Data Availability

The data supporting the findings of this study are available at the following Open Science Framework repository: https://osf.io/qvwb4/?view_only=0952d1b4b64c409daa4dbf120c6e1fab, accessed on 24 June 2025.

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
