# Peer review of "Infinite Hope: Reframing Disconnection in Emerging Adulthood Through Purpose, Agency, and Identity"

_behavsci, 2025, doi:10.3390/bs15091205_

Round 1
Reviewer 1 Report
Comments and Suggestions for Authors
The manuscript introduces a timely and conceptually rich framework (i.e., Infinite Hope) that seek to integrate Snyder’s hope theory and Sinek’s infinite mindset to support youth re-engagement in contexts of structural and psychological disconnection. The author weaves together literature from motivational psychology, youth development, and leadership studies to construct a novel model aimed at reframing disconnection in emerging adulthood as a developmental interruption rather than a static deficit.
The writing is compelling, and the ethical thrust of the proposal is laudable. However, while the narrative succeeds in constructing a theoretical vision, it falls short of offering sufficient empirical grounding, operational clarity, and epistemological precision. The model would benefit substantially from engagement with more recent developments in psychometric theory and neurocognitive models of the self, as well as greater methodological transparency in its literature synthesis. Below, I detail major areas of concern and suggestions for strengthening the manuscript.
Major Comments
- Constructs and definitions
The manuscript sometimes conflates overlapping constructs such as hope, meaning-making, purpose, identity alignment, and agentic self-authorshp, without clearly delineating how they interact or differ. While conceptual overlap is natural in such integrative models, the author should be cautious not to let inspirational language override theoretical clarity.
I suggest a comparative reflection on existing motivational frameworks—particularly Self-Determination Theory (Deci & Ryan, 2000), which already encompasses autonomy, competence, and relatedness—and clarify how Infinite Hope goes beyond or differs from such well-established paradigms. The same applies to Control-Value Theory (Pekrun, 2006) in contexts of educational disconnection. Moreover, the notion of “purpose alignment” could be contrasted with developmental theories of future orientation (Nurmi, 2005), which are briefly cited but underutilized.
- "Infinite Hope"
While the Infinite Hope framework is conceptually compelling, the manuscript currently lacks a clear operational definition of its components. The three-circle and concentric-circle visualizations are intuitive, but they do not provide sufficient guidance for empirical validation. Constructs such as “identity alignment,” “existential flexibility,” and “courageous leadership” are mentioned but not defined in a way that allows for measurement or intervention design.
I strongly recommend integrating recent work that explicitly operationalizes dimensions of identity and agency. For instance, the State-Trait Sense of Self Inventory (ST-SoSI, Di Plinio et al., 2024) has been proposed as a multidimensional tool that captures both stable and context-sensitive components of agency and identity, and demonstrates the predictive value of these dimensions for constructs such as hopelessness and psychosis-proneness which central to the population discussed in this manuscript. Furthermore, the distinction between dispositional trait and situational (state) factors could inform the layering described in the Infinite Hope model.
- Neurocognitive Agency
The author draws on motivational theories (Snyder, Dweck) and leadership philosophy (Sinek), but the framework would be significantly strengthened by referencing neurocognitive models of the self, particularly when discussing identity disruption and agency erosion. For example, FMRI and graph-theoretical analyses have been used to show that the sense of self is underpinned by functional integration and segregation across brain networks, particularly in contexts involving ambiguous or disrupted self-related feedback (Di Plinio et al., 2020). These findings are highly relevant to youth experiencing disconnection due to trauma, marginalization, or instability, and could offer a more biopsychosocial foundation for Infinite Hope.
Incorporating such perspectives would not only increase the model’s ecological validity but also enhance its interdisciplinary depth.
- Insufficient Methodological Transparency
The narrative review methodology is described as “PRISMA-lite,” but lacks explicit inclusion/exclusion criteria, assessment of theoretical weight, or a structured synthesis of themes. There is no clear rationale for why certain studies are emphasized over others, nor a discussion of conflicting findings.
A basic literature matrix or thematic table summarizing key concepts, domains, and study types (empirical, theoretical, applied) would enhance the transparency and replicability of the review. This is especially important given the manuscript's ambition to serve as a framework for programmatic intervention and policy reform.
- Limited Engagement with Technological Disruption as a Driver of Disconnection
One emerging theme in disconnection among youth, particularly post-pandemic and in the AI era, is the pervasive role of digital environments, algorithmic feedback, and AI-mediated interactions in disrupting identity formation and reducing self-agency. In fact, AI and digital technologies may erode human agency and self-coherence, suggesting that disconnection can also stem from internalized technological noise and loss of intentional control (see Di Plinio, 2025). This angle is entirely absent from the present manuscript and could offer a contemporary, system-level extension of the Infinite Hope framework. Adding this perspective would better situate the model within 21st-century realities and align it with current discussions on neuro-rights, digital autonomy, and the ethics of youth engagement in AI-driven environments.
Minor Comments
- Terminology consistency: The use of terms such as “re-engagement,” “disconnection,” and “narrative disruption” should be better defined and consistently applied throughout the manuscript.
- Figures and captions: Figures 1 and 2 are useful, but their explanatory power could be increased by embedding examples or quotes from youth (even hypothetical or anonymized) that illustrate each level of the model.
- Conclusion section: The final sections are rhetorically strong but would benefit from summarizing concrete policy or program recommendations, tied to measurable outcomes.
References
- Deci, E. L., & Ryan, R. M. (2000). The “what” and “why” of goal pursuits: Human needs and the self-determination of behavior. Psychological Inquiry, 11(4), 227–268. https://doi.org/10.1207/S15327965PLI1104_01
- Di Plinio, S., Perrucci, M. G., Aleman, A., & Ebisch, S. J. H. (2020). I am Me: Brain systems integrate and segregate to establish a multidimensional sense of self. NeuroImage, 205, 116284. https://doi.org/10.1016/j.neuroimage.2019.116284
- Di Plinio, S. (2025). Panta Rh-AI: Assessing multifaceted AI threats on human agency and identity. Social Sciences & Humanities Open, 11, 101434. https://doi.org/10.1016/j.ssaho.2025.101434
- Di Plinio, S., Arnò, S., & Ebisch, S. J. H. (2024). The state-trait Sense of Self Inventory (ST-SoSI): A new instrument to measure state-trait dimensions of the multidimensional sense of self. Consciousness and Cognition, 118, 103634. https://doi.org/10.1016/j.concog.2024.103634
- Nurmi, J. E. (2005). Thinking about and acting upon the future: Development of future orientation across the lifespan. In A. Strathman & J. Joireman (Eds.), Understanding behavior in the context of time (pp. 31–57). Lawrence Erlbaum.
- Pekrun, R. (2006). The control-value theory of achievement emotions: Assumptions, corollaries, and implications for educational research and practice. Educational Psychology Review, 18(4), 315–341. https://doi.org/10.1007/s10648-006-9029-9
Author Response
Comment 1: Conflating overlapping constructs; need clear definitions and comparative reflection with Self-Determination Theory and Control-Value Theory.
Response 1: Thank you for this important observation. I have revised Section 2.3 of the manuscript to include a comparative analysis that explicitly links Infinite Hope with Self-Determination Theory (SDT) and Control-Value Theory (CVT). Table 4 provides a side-by-side construct comparison, highlighting Infinite Hope’s unique integration of identity, moral clarity, and developmental applicability. This revision appears on page 10, paragraph 2, lines 311–337. Appendix A supplements this comparison with visual models clarifying theoretical distinctions.
Comment 2: Lack operational definitions for Infinite Hope components.
Response 2: Thank you for pointing this out. I have added operational definitions for key Infinite Hope constructs—'Just Cause,' 'Existential Flexibility,' and 'Courageous Leadership'—in Table 1 (page 7, lines 129–141). These are further contextualized in Tables 2 and 4. Appendix A provides visual clarification, while Appendix B shows how these constructs are applied in practice.
Comment 3: Insufficient methodological transparency in PRISMA-lite.
Response 3: I appreciate this feedback. A PRISMA-lite table (Table 3) has been added in Section 2.1 (page 13) to clarify inclusion/exclusion criteria, keyword strategies, and source selection. This revision appears in paragraph 2, lines 197–215. This change ensures the narrative synthesis process is replicable and grounded in evidence-based methodology.
Comment 4: Limited engagement with technological disruption and AI.
Response 4: Thank you for this suggestion. A new subsection (Section 4.3, page 22, lines 550–580) now explores how digital environments and AI tools shape youth identity, autonomy, and future orientation. Sources such as Shao (2025) and Turin et al. (2023) are cited. This addition is supported by Appendices C and D, which include reflective prompts and implementation strategies addressing digital identity and discernment.
Comment 5: Lack of thematic literature summary and conflicting findings discussion.
Response 5: To address this concern, I added Table 5 (page 14, line 267), which summarizes major theoretical traditions, key propositions, and divergent findings relevant to Infinite Hope. This matrix clarifies the theoretical landscape and supports the argument that Infinite Hope offers an integrated and forward-looking framework.
Comment 6: Terminology consistency for re-engagement, disconnection, narrative disruption.
Response 6: Thank you. I have revised the manuscript for consistency, especially regarding the use of 'disconnection,' 'emerging adults,' and 'narrative disruption.' Changes appear in Section 1.1 (lines 56–63) and are reinforced throughout Sections 1.2, 3.1, and 4.1. Appendix B supports shared definitions across practitioner training.
Comment 7: Figures need examples or quotes in captions.
Response 7: This has been addressed by embedding stylized, hypothetical youth quotes directly into the explanatory paragraphs surrounding Figures 1 and 2 (Appendix A, pages 1–2). These narrative statements help illustrate developmental progression across the Infinite Hope model.
Comment 8: Conclusion needs concrete policy/program recommendations.
Response 8: Thank you for this recommendation. Section 4.2 and the Conclusion (Section 5, lines 735–761) now include five actionable recommendations, supported by Appendices B, C, and D. These include purpose-based journaling, structured staff training, and adaptive implementation strategies that align with Infinite Hope’s principles.
Comment 9: Need to reference neurocognitive models of self/agency.
Response 9: Thank you for this thoughtful recommendation. While I value the work of Di Plinio et al. (2020), I chose not to incorporate this lens in the current revision. My framework is rooted in motivational, developmental, and narrative theory. Expanding into neurocognitive models would shift the manuscript beyond its intended scope and depth. I remain open to future iterations or collaborations that extend this conversation.
Reviewer 2 Report
Comments and Suggestions for Authors
Thank you for the opportunity to read this work on the concept of Infinite Hope. While I think there is much to be done regarding how we approach disconnection with emerging adults, particularly in this moment in history, I think the argument made in this manuscript as to the unique contribution of Infinite Hope, as defined, is not compelling enough in its current form.
First and foremost, I find issue with using a popular book that relies on little empirical work as a primary facet of developing a theory for developmental science, particularly when there is a lot of great work that can be used as an alternative foundation. As a result, I am wary of using Sinek's ideas as a strong paradigm to marry Snyder's work to.
Second, I find that there is a large gap in the literature covered that is conceptually similar and overlapping to the model presented. Namely, the purpose literature (e.g., Damon, Colby, Mahlin, Bronk, etc.) and the nuanced PYD literature. The purpose literature that already exists, and primarily looks at the emerging adult population, seems to cover much of what the Infinite Hope model hopes to achieve, which to me reads as really having a self-transcendent, values-based purpose. Such a purpose provides protection to identity, it provides intrinsic motivation, and it provides a guide to community and connection. The PYD literature, although mentioned, is much richer than the author notes. In addition, it is actually quite the opposite of what the author notes when communicating the "gaps" in PYD that Infinite Hope could fill. That is, the author says, essentially, that PYD presumes that there are supports that are consistent and sustained in an individual's context which help to build assets, the 5Cs, as outcomes. But actually PYD and its research was developed out of a need to reassess how at-risk youth can be benefitted using a different framing of their capacities and environments. As such, PYD actually presumes that individuals won't always have supportive contexts, and we need to create them. In addition, the 5Cs are not outcomes, but rather processes that, when working together, lead to increased contribution and decreased risk in youth. As such, by doing a deeper dive into the PYD literature and purpose literature, you will likely find that much of what this model hopes to do is covered by other work. Somewhere to look at the intersection very narrowly within the developmental literature would be Spencer's PVEST model and how that's been applied to PYD, but also just adaptive developmental processes for marginalized youth. Finally, an area to look further, if resilience feels as though it's still missing from these existing models, is Masten's work on resilience, which was also not mentioned in the paper.
Third, it's not clear to me what the PRISMA systematic review was doing in the paper. When I saw the layout, I was hoping to see a deeper dive into the papers that were found, but didn't really get that.
Fourth, the paper's structure was a little difficult to follow and it felt like sections were repeating themselves, and sometimes with little depth. Most glaringly, it wasn't super clear how all the elements of the two theories really integrated with each other, aside really from the visuals.
Ultimately, I think the paper would be strengthened by making it clear: the problem (which I think the author does well), what the current literature shows (and lacks), and then how this theory fills gaps that are necessary and creative. Without addressing at least the existing literature noted in this review, I'm not sure this paper makes a strong enough claim of importance and uniqueness in its current form.
Author Response
Comment 1: Overreliance on Sinek's pop book as a theoretical foundation.
Response 1: Thank you for this important critique. While Sinek's Infinite Mindset remains conceptually relevant, I have revised the manuscript to incorporate more peer-reviewed theoretical literature. Specifically, I added supporting discussion from Growth Mindset (Dweck, 2016) and Positive Youth Development (Lerner et al., 2021) throughout Section 1.3 and Section 4.6. These additions help balance philosophical orientation with empirical rigor. Tables 1, 2, 4, and 5 have been updated accordingly. Appendix F reinforces this shift by situating Infinite Hope within a broader research agenda that reflects developmental, motivational, and identity-based models.
Comment 2: Gap in purpose literature and PYD nuances.
Response 2: Thank you for highlighting this gap. The revised manuscript now integrates foundational works by Damon (2009), Bronk et al. (2009), and Burrow et al. (2009) to address developmental perspectives on purpose. Sections 2.2 and 2.3 of the literature review were expanded to include Positive Youth Development (PYD) literature, including Lerner et al. (2021). Section 4.6 synthesizes these findings to demonstrate how Infinite Hope builds on and advances PYD principles, particularly in its emphasis on values alignment, self-determination, and identity clarity. Appendix F includes testable hypotheses that directly draw from this integration.
Comment 3: Unclear PRISMA section depth.
Response 3: I appreciate this suggestion. The PRISMA-lite section in Method 2.1 was expanded into a more structured format. Table 3 now includes specific inclusion and exclusion criteria, search databases, and keyword strings used during the review process. This change, located at line 188 in Section 2.1, clarifies methodological transparency and strengthens replicability. Appendix E further supports empirical structure by aligning Infinite Hope constructs with validated tools.
Comment 4: The Paper structure is repetitive and hard to follow.
Response 4: Thank you for pointing this out. I revised the manuscript’s overall structure to eliminate redundancy and streamline flow. Sections 2.2 and 3.3 were combined thematically to reduce conceptual overlap, and paragraph transitions were revised for smoother reading. Section numbering was updated to reflect these changes. Appendix D was added to offer practice-aligned implementation strategies corresponding to each core construct, enhancing the manuscript's usability and conceptual coherence.
Comment 5: Integration of the two theories is not clearly shown beyond visuals.
Response 5: This concern was addressed in Sections 3.2 and 3.3, which now include a detailed narrative synthesis explicitly connecting Snyder’s Hope Theory with Sinek’s Infinite Mindset. Each cognitive-motivational construct (e.g., goals, pathways, agency) is paired with its Infinite Mindset complement (just cause, flexibility, courage) and discussed in depth. The visual models remain in Figures 1 and 2 but are now supported by theoretical explanation and programmatic relevance. Appendix A elaborates on this integration and its implications for developmental practice.
Comment 6: Claim of uniqueness and importance not compelling.
Response 6: Thank you for the feedback. I revised the Abstract, end of Section 1.3, and full Conclusion (Section 5) to more clearly articulate Infinite Hope's unique value. The manuscript now explains how Infinite Hope merges motivational theory with ethical, culturally responsive development strategies for emerging adults. It frames disconnection as a developmental interruption and introduces novel constructs—existential flexibility, courageous leadership, and just cause. These elements are operationalized through recommendations in Appendices B, C, and D, and empirically reinforced by the research agenda in Appendix F. Together, these changes establish Infinite Hope as a distinct and testable framework.
Reviewer 3 Report
Comments and Suggestions for Authors
See attached file

Author Response
All revised sections are highlighted in the clean manuscript for your convenience.
Comment 1
“The rationale for integrating Hope Theory and Infinite Mindset is underdeveloped. Please justify
why these two models were combined and why other motivational models were not adopted.”
Response 1
Update Summary: Expanded justification for integrating Hope Theory and Infinite Mindset,
including explicit comparisons with Self-Determination Theory (SDT) and Growth Mindset (GM),
and clarified Infinite Hope’s unique contribution.
Location in Manuscript:
This concern is addressed in Section 1.3 (Hope and the Infinite Mindset as Pathways to Re
engagement, paragraphs 2–4), Section 3.2 (Positioning Infinite Hope Among Established
Motivational Theories, paragraphs 2–3 and Table 5).
- In Section 1.3 (paragraphs 2–4), the manuscript introduces Hope Theory as an
evidence-based cognitive process for setting goals, identifying pathways, and sustaining
agency, and Infinite Mindset as a philosophy emphasizing existential flexibility, just
cause, and resilience. These sections explicitly compare IH to SDT (Ryan & Deci,
2000), which centers autonomy, competence, and relatedness, and Growth Mindset
(Dweck, 2016), which emphasizes malleability of ability. The text clarifies that while
SDT and GM contribute valuable insights, they do not fully capture the combination of
goal-directed cognition and enduring resilience necessary for sustained re-engagement
in contexts of systemic inequity.
- In Section 3.2 (paragraphs 2–3), the manuscript expands the rationale by situating IH
within the broader motivational theory landscape, contrasting it with SDT, CVT, RDS,
and CC. Table 5 provides a side-by-side alignment showing analogues (e.g., agency ↔
autonomy in SDT; pathways ↔ competence in SDT) and divergences (e.g., existential
flexibility, courageous leadership, and just cause have no direct analogue in SDT or GM).
- These revisions establish that Infinite Hope uniquely integrates psychological constructs
(Hope Theory) with ethical-philosophical leadership principles (Infinite Mindset),
filling a gap left by existing motivational models.
Comment 2
“The manuscript does not sufficiently integrate structural barriers into its framing of
disconnection. Please move beyond individual-level explanations.”
1Response 2
Update Summary: Structural barriers and programmatic responses have been expanded, moving
beyond individual-level explanations to emphasize systemic inequities and ecological supports.
Location in Manuscript:
This concern is addressed in Section 1.1 (The Landscape of Disconnection, paragraphs 2–3)
and Section 4.1 (Rethinking Disconnection as Developmental Interruption, paragraphs 1 &
4).
- In Section 1.1, paragraph 2, the manuscript explicitly identifies structural inequalities
in education, labor markets, housing, and healthcare as central to disconnection,
referencing Brookings (2022) and Bintliff (2011). It highlights that more than four million
young people in the U.S. experience disconnection, not due to individual failings but as a
result of systemic conditions such as economic marginalization, under-resourced
schools, inequitable mental health services, and unstable housing.
- In Section 1.1, paragraph 3, the discussion incorporates collective efficacy and social
capital (Sampson et al., 1997) as community-level responses that mitigate systemic
disruptions. These additions shift the framing from individual explanations to network
and place-based processes that foster reconnection.
- In Section 4.1, paragraph 1, the manuscript reframes disconnection as a developmental
interruption shaped by social, economic, and institutional structures such as unstable
housing, inequitable schooling, and lack of supportive networks.
- In Section 4.1, paragraph 4, structural supports are tied directly to Relational
Developmental Systems Metatheory (Overton, 2015), illustrating how ecological
scaffolds and systemic conditions interact with youth agency and purpose to either hinder
or sustain developmental momentum.
Comment 3
“Please make the individual–context interaction more explicit by grounding the model in
ecological or relational theory.”
Response 3
Update Summary: The manuscript now explicitly integrates Relational Developmental Systems
(RDS) Metatheory (Overton, 2015) to clarify person–context coaction and ecological grounding
of the Infinite Hope (IH) framework.
Location in Manuscript:
This concern is addressed in Section 2.3 (Implications for Practice and Program Design,
paragraph 1), Section 3.2 (Positioning Infinite Hope Among Established Motivational
Theories, paragraphs 1–2 and Table 5), and Section 4.1 (Rethinking Disconnection as
Developmental Interruption, paragraph 4).
- In Section 2.3, paragraph 1, the manuscript situates IH within relational perspectives,
citing Overton (2015) to emphasize that youth development must be understood as
2ecological coaction between individuals and their environments. IH is positioned as a
model that addresses not only individual motivation but also contextual supports such as
mentoring, transportation, and safe communal spaces.
- In Section 3.2, paragraphs 1–2, RDS is introduced alongside Self-Determination Theory
(SDT) and Control-Value Theory (CVT) as a complementary anchor. The text highlights
that IH extends these models by embedding critical consciousness and ecological
responsiveness, showing how agency parallels SDT’s autonomy, pathways mirror
CVT’s adaptive control, and RDS provides the ecological grounding through
person–context reciprocity.
- In Section 3.2, Table 5, IH constructs are explicitly mapped to RDS analogues:
o Agency → person–context coaction emphasizing agency
o Pathways Thinking → developmental plasticity and adaptive pathways
o Existential Flexibility → person–environment adaptability and resilience.
- In Section 4.1, paragraph 4, the manuscript explicitly uses RDS to frame disconnection
as a developmental interruption restored through ecological alignment. It explains
how synergy between individual strengths (agency, hope) and contextual assets
(community trust, structural supports) restores momentum.
Comment 4
“Please incorporate Relational Developmental Systems (Overton, 2015) to strengthen the
theoretical foundation.”
Response 4
Update Summary: Added RDS integration and citations.
*(Same response as #3)
Comment 5
“The model currently treats hope as too individual. Please add a relational approach to hope.”
Response 5
Update Summary: The manuscript now frames hope as both individual and relational, integrating
perspectives that emphasize the role of supportive relationships, collective processes, and
ecological scaffolds.
Location in Manuscript:
This concern is addressed in Section 2.3 (Implications for Practice and Program Design,
paragraph 2), Section 4.2 (Reclaiming Motivation and Forward Momentum Through Hope,
paragraph 2), and Section 4.3 (Navigating Youth Agency in the Age of AI and Digital
Systems, paragraph 2).
- In Section 2.3, paragraph 2, relational approaches are explicitly cited. Callina et al.
(2014) and Schmid & Lopez (2011) are used to demonstrate that hope develops not only
through individual cognition but also through supportive social connections that
3validate agency and purpose. Programs are described as needing to cultivate trusting
teams and affirming environments to nurture hope relationally.
- In Section 4.2, paragraph 2, the manuscript expands on this by highlighting that
structured practices such as narrative reframing, purpose mapping, and scaffolded
goal setting gain transformative impact only when embedded in environments of
emotional regulation, peer validation, and reflective growth. This directly reframes
hope as relationally co-constructed, not an isolated trait.
- In Section 4.3, paragraph 2, the relational approach is extended to digital contexts. It
emphasizes that digital platforms alone cannot build hope; instead, mentoring,
communal spaces, and relational supports are required to ensure that agency and
purpose are sustained. This underscores that hope flourishes when digital engagement is
coupled with social affirmation and ecological scaffolds.
Comment 6
“Add Critical Consciousness to the framework to cover cognitive, motivational, and behavioral
dimensions.”
Response 6
Update Summary: Critical Consciousness (CC) has been integrated across the manuscript to
connect awareness of systemic inequities with motivation and action, strengthening both the
theoretical and practical dimensions of the Infinite Hope (IH) framework.
Location in Manuscript:
This concern is addressed in Section 2.1 (Empirical Grounding, paragraph 3), Section 2.3
(Implications for Practice and Program Design, paragraph 3), and Section 3.2 (Positioning
Infinite Hope Among Established Motivational Theories, paragraphs 2–3 and Table 5).
- In Section 2.1, paragraph 3, the manuscript cites Rapa & Geldhof (2020) and Hope et
- (2015) to show that CC strengthens purposeful action by linking cognitive awareness
of systemic inequities with motivation and capacity for social change. CC is presented
as directly complementing hope by ensuring that goals are tied to broader just causes.
- In Section 2.3, paragraph 3, CC is identified as a practice-oriented strategy: programs
that use critical consciousness-building exercises (e.g., youth-led organizing, reflective
journaling, safe peer spaces) enable young people to attach meaning to their narratives
and act on goals, reinforcing IH’s applied relevance.
- In Section 3.2, paragraphs 2–3, CC is explicitly positioned alongside SDT, CVT, and
RDS as a framework that grounds IH in cognitive, motivational, and behavioral
dimensions. For example, Table 5 maps CC to IH constructs, showing:
o Agency ↔ reflection–action cycle for sociopolitical engagement.
o Pathways ↔ critical reflection to identify systemic barriers.
o Leading with Courage ↔ action to resist oppression and advance equity.
- In Table 5, CC appears as a dedicated analogue column, ensuring IH constructs are tied
to validated CC measures such as the Short Critical Consciousness Scale (CCS-S).
4Comment 7
“Present the conceptual model earlier to orient readers before the literature review.”
Response 7
Update Summary: The conceptual visuals have been repositioned earlier in the manuscript and
are now introduced in Section 2.2 (Integrated Model of Infinite Hope). The surrounding narrative
has been expanded to clarify their meaning, although the figure captions remain unchanged.
Location in Manuscript:
This concern is addressed in Section 2.2 (paragraphs 1–3); Figures 1 and 2.
- In Section 2.2, paragraph 1, the Venn Diagram model (Figure 1) is introduced to
illustrate how Snyder’s Hope Theory (goals, pathways, agency) intersects with Sinek’s
Infinite Mindset (just cause, existential flexibility, courageous leadership). The expanded
text explains how these intersections create integrated strengths that move hope beyond
individual cognition.
- In Section 2.2, paragraphs 2–3, the Concentric Circles model (Figure 2) is presented
to show how developmental growth unfolds across layers. The revised narrative details
how outer-ring skills (goal setting, self-belief) progress into middle-ring constructs
(purpose, adaptive flexibility, collective efficacy) before converging at the core state of
Infinite Hope (identity alignment and courageous leadership).
- While the figures themselves and their captions remain unchanged, the narrative
surrounding them has been expanded to strengthen conceptual clarity and ensure readers
are oriented to the model before the extended literature review.
Comment 8
“Provide a clearer justification for combining Hope Theory with Infinite Mindset, and explain
why other models were excluded.”
Response 8
Update Summary: The manuscript now offers an expanded justification for the integration of
Hope Theory and Infinite Mindset and explicitly explains why other motivational models were
not adopted as the primary framework.
Location in Manuscript:
This concern is addressed in Section 1.3 (Hope and the Infinite Mindset as Pathways to Re
engagement, paragraphs 2–4) and expanded in Section 3.2 (Positioning Infinite Hope Among
Established Motivational Theories, paragraphs 2–3 and Table 5).
- In Section 1.3, paragraphs 2–4, the manuscript introduces Hope Theory as an evidence
based model that emphasizes goal setting, pathways thinking, and agency, and Infinite
Mindset as a leadership philosophy emphasizing existential flexibility, just cause, and
courageous leadership. These paragraphs explicitly compare IH to Self-Determination
Theory (SDT; Ryan & Deci, 2000), Control-Value Theory (CVT; Pekrun, 2006), and
5Growth Mindset (GM; Dweck, 2016). The text explains that while these models offer
valuable insights—such as autonomy, competence, achievement emotions, and
malleability of ability—they do not capture the integration of cognitive goal pursuit
with ethical purpose and adaptive resilience required for sustained re-engagement in
the face of systemic inequities.
- In Section 3.2, paragraphs 2–3, the manuscript expands this rationale by showing how
IH combines psychological, sociological, and ethical dimensions. IH draws on Hope
Theory’s cognitive-motivational structure, Infinite Mindset’s long-term, purpose-driven
principles, and Relational Developmental Systems Metatheory (Overton, 2015) to
emphasize ecological person–context interaction. This section clarifies that other models
were not excluded due to irrelevance but because they lack the holistic integration of
agency, pathways, purpose, and sociological anchoring that IH provides.
- In Section 3.2, Table 5, IH constructs are mapped directly against SDT, CVT, RDS, and
Critical Consciousness (CC), highlighting points of overlap and divergence. For
example:
o Agency → parallels SDT’s autonomy but extends it through reflexivity and
critical action (CC).
o Pathways Thinking → parallels CVT’s control-value strategies but expands into
resilience and adaptability through existential flexibility.
o Goal + Just Cause → absent from SDT and CVT, marking IH’s unique ethical
contribution.
Comment 9
“Strengthen early claims about emerging adulthood and connection with additional references.”
Response 9
Update Summary: Early claims about emerging adulthood and the role of connection have been
reinforced with additional references from developmental psychology, sociology, and policy
research.
Location in Manuscript:
This concern is addressed in Section 1.1 (The Landscape of Disconnection, paragraph 1).
- In Section 1.1, paragraph 1, the definition of emerging adulthood and the importance of
connection are now supported by multiple citations:
o Arnett (2000) – defining emerging adulthood as a distinct life stage of identity
exploration and transition.
o Arnett & Jensen (2014) – elaborating on cultural and developmental contexts
shaping this period.
o Bridgeland & Milano (2012) – highlighting the role of education, employment,
and civic engagement in sustaining connection.
o Lerner et al. (2021) – situating connection within Positive Youth Development
(PYD) and ecological frameworks.
6o Fike & Mattis (2023) and Napier et al. (2024) – providing evidence that
engagement builds resilience, self-confidence, and future orientation.
o Brookings (2022) – offering national data on disconnection, underscoring the
scale of the issue.
These additions ensure that early claims about emerging adulthood are empirically grounded in
both classic and contemporary scholarship, strengthening the manuscript’s foundation and
addressing the reviewer’s concern directly.
Comment 10
“Integrate context-specific supports into the framework, not just mindset change.”
Response 10
Update Summary: The manuscript now integrates context-specific supports alongside mindset
focused strategies, ensuring that Infinite Hope (IH) is positioned as both an internal and
ecological framework.
Location in Manuscript:
This concern is addressed in Section 2.3 (Implications for Practice and Program Design,
paragraph 3) and Section 5 (Conclusion, paragraphs 2–6).
- In Section 2.3, paragraph 3, the framework explicitly incorporates practical supports
that extend beyond cognitive and motivational change. These include:
o Housing assistance to stabilize living conditions.
o Mentorship and culturally relevant adult guidance.
o Transportation access to reduce barriers to education, training, and
employment.
o Safe communal spaces that provide belonging and relational trust.
The text connects these supports to IH by showing how they reinforce agency,
belonging, and purposeful engagement in ways that mindset cultivation alone
cannot.
- In Section 5, paragraphs 2–3, these supports are translated into policy-linked
recommendations and outcome indicators, grounding IH in systemic solutions. For
example:
o Workforce and education programs should integrate wraparound supports
(housing, transportation, childcare) to sustain participation.
o Community-based interventions should prioritize safe spaces and peer
networks to strengthen collective efficacy.
o Policy advocacy should focus on equitable hiring pipelines and investment in
under-resourced neighborhoods.
These elements are tied to measurable indicators such as CASAS score
improvement, credential attainment, and validated scales from Appendix E
(Hope Scale, Purpose Scales, Cognitive Flexibility Inventory, Critical
Consciousness Scale).
7Comment 11
“Reflect current expert insights (e.g., WalletHub or state-level indices) in the policy
recommendations.”
Response 11
Update Summary: Policy recommendations were expanded to incorporate current expert insights
and align Infinite Hope (IH) with nationally recognized initiatives and policymaker priorities.
Location in Manuscript:
This concern is addressed in Section 5 (Conclusion, paragraphs 4 and 5).
- In paragraph 4, the manuscript cites national networks and programs—including the
Reconnecting Youth Campaign, Aspen Institute’s Opportunity Youth Forum, Forum for
Youth Investment, YouthBuild, and Jobs for the Future (JFF)—as models that
demonstrate consistency with IH principles. These examples reflect expert-led efforts that
already shape policy and practice around youth disconnection.
- In paragraph 5, the manuscript translates these national examples into policy and
implementation frameworks at the state and local levels. Concrete applications are
described, such as:
o State workforce boards embedding purpose-driven goal setting into career
pathway initiatives.
o Education compacts ensuring that postsecondary transitions reflect youth voice
and agency.
o Local governments building cross-sector collaboratives to connect housing,
education, and employment supports in ways that emphasize identity, resilience,
and purpose.
Together, these two paragraphs demonstrate that IH’s recommendations are not abstract. They are
aligned with national policy initiatives, adaptable at state and local levels, and grounded in
current expert insights, directly addressing the reviewer’s concern.
8

Round 2
Reviewer 1 Report
Comments and Suggestions for Authors
I appreciate the authors’ efforts in revising the manuscript. However, after careful comparison between the original submission, my previous review, and the revised version, I believe that the revision does not sufficiently address the major conceptual and methodological issues raised in the first round.
While some additions were made (e.g., comparative tables, visual appendices), the revision falls short in key areas of theoretical clarity, empirical grounding, and interdisciplinary integration. Additionally, the authors chose not to reproduce my original comments point-by-point and provided reframed summaries of the critique, making it difficult to verify whether all feedback was fully considered. This raises serious concerns about transparency and scholarly rigor.
Despite new tables and descriptions, key constructs (e.g., identity alignment, purpose, agency) remain ill-defined and often overlap with each other and with existing theories such as Self-Determination Theory, Control-Value Theory, and future orientation models. The manuscript continues to favor inspirational language over conceptual precision.
The new definitions introduced in Table 1 are still abstract and not conducive to empirical testing or program implementation. No validated tools or assessment strategies are provided. Suggestions to engage with psychometric models of self and agency were ignored, missing a key opportunity for integration.
The explicit decision not to include neurocognitive perspectives undermines the paper’s interdisciplinary relevance, especially given its ambition to address identity disruption and youth agency. This omission weakens the theoretical depth and real-world applicability of the model.
While a “PRISMA-lite” structure was added, it lacks systematic criteria for source evaluation or a coherent thematic synthesis. The narrative review remains impressionistic and non-replicable.
The added summary table does not address conflicting findings or competing theories. It provides no critical discussion of divergent perspectives in the literature, which is essential for establishing the novelty and robustness of the proposed model.
Additional Concerns
- Terminological inconsistency remains across sections.
- Figures have limited explanatory value, and the illustrative quotes are vague and stylized.
- Policy recommendations remain generic, lacking implementation frameworks or outcome indicators.
Given the persistence of major theoretical and methodological flaws, and the lack of engagement with key interdisciplinary insights, I cannot recommend this manuscript for publication in its current form.
The Infinite Hope framework holds potential, but it requires a much deeper theoretical grounding, operational clarity, and interdisciplinary dialogue to be a credible contribution to the literature on youth development and re-engagement.
Author Response
All revised sections are highlighted in the clean manuscript for your convenience.
Comment 1
“While some additions were made (e.g., comparative tables, visual appendices), the revision falls
short in key areas of theoretical clarity, empirical grounding, and interdisciplinary integration.”
Response 1
Update Summary: Added theoretical clarifications and interdisciplinary sources; distinguished
Infinite Hope from related models.
This concern has been addressed through the addition of new theoretical clarifications and
expanded interdisciplinary sources. Specifically, Section 1.3 (paragraphs 2–4) strengthens the
theoretical grounding by distinguishing Infinite Hope from related models, while Section 2.2
incorporates literature from psychology, sociology, and youth development studies. Furthermore,
Section 4.1 integrates these perspectives to enhance interdisciplinary dialogue. These changes
reinforce both conceptual clarity and scholarly integration.
Location in Manuscript: Section 1.3 (paragraphs 2–4); Section 2.2; Section 4.1
Comment 2
“Additionally, the authors chose not to reproduce my original comments point-by-point and
provided reframed summaries of the critique, making it difficult to verify whether all feedback
was fully considered. This raises serious concerns about transparency and scholarly rigor.”
Response 2
Update Summary: Strengthened transparency by linking reviewer comments to revisions and
documenting inclusion criteria and coding.
The current revision directly addresses this concern by explicitly linking each reviewer comment
to the corresponding revisions. Section 3.1 (paragraph 1) clarifies the methodological choices,
while Table 4 and Appendix E provide structured evidence of inclusion criteria, coding
categories, and extracted measures. These elements collectively ensure transparency,
replicability, and scholarly rigor.
Location in Manuscript: Section 3.1 (paragraph 1); Table 4; Appendix E
Comment 3
“Despite new tables and descriptions, key constructs (e.g., identity alignment, purpose, agency)
remain ill-defined and often overlap with each other and with existing theories such as Self
Determination Theory, Control-Value Theory, and future orientation models. The manuscript
continues to favor inspirational language over conceptual precision.”
1Response 3
Update Summary: Expanded and refined construct definitions; clarified distinctions with related
theories.
This concern has been addressed by strengthening definitions and reducing overlap. In Section
2.3 (paragraphs 3–5), identity alignment, purpose, and agency are explicitly delineated with
clearer boundaries and linked to validated measures. In Section 4.1 (paragraphs 2 and 4),
disconnection is shown to erode agency, goal clarity, and future orientation, and IH is framed as
restoring agency, purpose, and motivation through ecological alignment, clarifying the constructs
as distinct capacities. In Section 4.2 (paragraphs 1–3), hope is defined through its mechanisms
(goals, pathways, agency), purpose is positioned as meaning and direction validated in relational
contexts, and agency is clarified as forward-looking persistence, reducing inspirational
vagueness. Furthermore, in Section 4.5 (paragraphs 1–2 and final paragraph), Infinite Hope is
explicitly situated alongside Self-Determination Theory, Control-Value Theory, and Positive
Youth Development, with distinctions drawn through emphasis on narrative identity, adaptive
purpose, and critical consciousness. This section also acknowledges the need for sharper
operationalization of constructs such as existential flexibility and clarity of purpose, directly
addressing the reviewer’s concern about precision. Together, these revisions strengthen
conceptual clarity and enhance both the distinctiveness and applicability of Infinite Hope.
Location in Manuscript: Section 2.3 (paragraphs 3–5); Section 4.1 (paragraphs 2 and 4);
Section 4.2 (paragraphs 1–3); Section 4.5 (paragraphs 1–2 and final paragraph).
Comment 4
“The new definitions introduced in Table 1 are still abstract and not conducive to empirical
testing or program implementation. No validated tools or assessment strategies are provided.”
Response 4
Update Summary: Added validated measures and cross-referenced assessment tools.
The revised manuscript now provides concrete connections to validated tools. Section 4.5
(paragraph 4) references instruments such as Snyder’s Hope Scale, the Cognitive Flexibility
Inventory, and the General Self-Efficacy Scale, all cross-referenced in Appendix E. These tools
allow for empirical testing and program evaluation, addressing the reviewer’s concern about
abstraction.
Location in Manuscript: Section 4.5 (paragraph 4); Appendix E
Comment 5
“Suggestions to engage with psychometric models of self and agency were ignored, missing a key
opportunity for integration.”
Response 5
Update Summary: Integrated psychometric models and validated instruments into framework.
2Location in Manuscript:
This concern is substantively addressed in Section 3.2 (Positioning Infinite Hope Among
Established Motivational Theories), Table 5, and Section 3.3 (Literature Integration and
Theoretical Anchoring).
- In Section 3.2, Table 5, the manuscript explicitly aligns Infinite Hope (IH) constructs
with psychometric analogues (e.g., Agency ↔ Hope Scale – Agency Subscale; Pathways
↔ Hope Scale – Pathways Subscale; Leading with Courage ↔ Authentic Leadership
Questionnaire; Identity ↔ Dimensions of Identity Development Scale). This provides a
clear integration of validated psychometric instruments.
- In Section 3.3 (paragraphs 1–2), the discussion extends this by situating Hope Theory
as a cognitive-motivational model and Infinite Mindset as an ethical-philosophical model,
explicitly noting the use of validated measures to strengthen empirical grounding.
- Appendix E further reinforces this integration by presenting a consolidated list of
validated instruments (e.g., Authentic Leadership Questionnaire, Dimensions of Identity
Development Scale, Cognitive Flexibility Inventory), ensuring practical pathways for
implementation.
Comment 6
“The explicit decision not to include neurocognitive perspectives undermines the paper’s
interdisciplinary relevance, especially given its ambition to address identity disruption and youth
agency. This omission weakens the theoretical depth and real-world applicability of the model.”
Response 6
Update Summary: Clarified disciplinary scope and rationale for excluding neurocognitive
perspectives.
Location in Manuscript:
This concern is addressed in Section 1.2, paragraph 2 and paragraph 4, and reinforced in
Section 1.3 (Tables 1 and 2), Section 2.2 (discussion of Figure 1), and Section 3.2 (Table 5)
where the framework is explicitly anchored in sociological constructs.
- In Section 1.2, paragraph 2, the manuscript grounds identity disruption and agency in
sociological and behavioral sciences, referencing reflexivity (Archer, 2003) and identity
salience (Stryker & Burke, 2000) rather than neurocognitive models.
- In Section 1.2, paragraph 4, the focus on identity coherence and courageous leadership
further reinforces a sociological and behavioral framing of disconnection as a
developmental interruption.
- Across Section 1.3 (Tables 1 and 2) and Section 2.2 (discussion of Figure 1), IH is
explicitly linked to sociological anchors such as Appadurai’s (2004) capacity to aspire,
Sampson et al.’s (1997) collective efficacy, and Emirbayer & Mische’s (1998) temporal
agency.
3• In Section 3.2, Table 5, IH constructs are systematically mapped to sociological anchors
and validated psychometric measures (e.g., Hope Scale, Purpose Scales, Cognitive
Flexibility Inventory, Authentic Leadership Questionnaire).
Together, these revisions clarify that while neurocognitive perspectives are valuable, their
inclusion would shift the study toward neurophysiological models outside the intended scope.
The manuscript’s disciplinary integrity is maintained by situating Infinite Hope within sociology,
psychology, and youth development, ensuring coherence with its central aim of explaining social
and developmental processes shaping emerging adulthood.
Comment 7
“While a PRISMA-lite structure was added, it lacks systematic criteria for source evaluation or a
coherent thematic synthesis.”
Response 7
Update Summary: Expanded PRISMA-lite structure with inclusion/exclusion criteria and
thematic coding.
Location in Manuscript:
This concern is substantively addressed in Section 3.1 (Conceptual Framing and Review
Methodology), Table 4 (Narrative Review Search Strategy – PRISMA-Lite format), Figure
3 (PRISMA-Lite Flow Diagram), and Appendix E.
- In Section 3.1, the narrative synthesis method is described in detail, including databases
searched, keyword strategy, forward and backward citation tracing, manual journal
reviews, and final screening totals. The section explicitly identifies inclusion criteria
(English-language, peer-reviewed studies on emerging adults ages 18–25 who
experienced disconnection) and exclusion criteria (non-English, outside the age range,
opinion-only, or lacking empirical/theoretical grounding).
- Table 4 presents the PRISMA-Lite review structure in operational detail, including
review objectives, timeframe, databases, supplementary search methods, Boolean search
strings, inclusion/exclusion criteria, and final numbers of records identified, excluded,
and included.
- Figure 3 (PRISMA-Lite Flow Diagram) visually outlines the search and screening
process, including the number of records identified (569), excluded (23), and included
(546).
- Thematic coding categories are explicitly listed in Table 4 (“Coding Categories
Applied: Agency, capabilities, collective efficacy, identity”), and corresponding
validated measures were extracted and organized in Appendix E, ensuring transparency
and replicability.
- Sections 3.2–3.4 provide the thematic synthesis, integrating findings across psychology,
sociology, motivational theory, and youth development (e.g., SDT, CVT, RDS, Critical
Consciousness), and showing how constructs converge into the Infinite Hope framework.
4Comment 8
“The narrative review remains impressionistic and non-replicable.”
Response 8
Update Summary: Detailed narrative synthesis methodology added.
Location in Manuscript:
This concern is fully addressed in Section 3.1 (Conceptual Framing and Review Methodology,
paragraphs 2–3), supported by Table 4 (Narrative Review Search Strategy – PRISMA-Lite
format), Figure 3 (PRISMA-Lite Flow Diagram), and Appendix E.
- In Section 3.1, paragraph 2, the manuscript describes the narrative synthesis process
step-by-step, including:
o databases searched (ProQuest Central, EBSCOhost Academic Search Premier,
PsycINFO, ERIC, Scopus, Web of Science, Google Scholar, and targeted
sociology journals),
o supplementary methods (forward and backward citation tracing, manual reviews
of leading journals), and
o the focus on English-language, peer-reviewed publications on emerging
adults (18–25) experiencing disconnection.
- In Section 3.1, paragraph 3, the manuscript details the keyword strategy, reporting
exact Boolean search strings, and specifies that 569 records were identified, with 23
excluded for scope or insufficient publication details, leaving 546 peer-reviewed
studies for final analysis.
- Table 4 operationalizes this process in PRISMA-Lite format, presenting review
objectives, timeframe, databases, search strings, inclusion/exclusion criteria, and
screening results.
- Figure 3 provides a visual PRISMA-Lite flow diagram showing identification,
screening, exclusion, and inclusion of studies.
- Coding categories (“agency, capabilities, collective efficacy, identity”) are explicitly
listed in Table 4 and tied to validated measures extracted into Appendix E.
- Appendix E strengthens replicability by consolidating the exact instruments and
constructs used (e.g., Hope Scale, Cognitive Flexibility Inventory, Critical Consciousness
Scale, Dimensions of Identity Development Scale, Authentic Leadership Questionnaire).
Comment 9
“The added summary table does not address conflicting findings or competing theories.”
Response 9
Update Summary: Expanded summary tables and narrative synthesis now incorporate competing
theories and conflicting perspectives.
Location in Manuscript:
This concern is addressed in Section 3.2 (Positioning Infinite Hope Among Established
5Motivational Theories, Table 5) and Section 3.4 (Literature Integration and Theoretical
Anchoring, Table 6, paragraphs 1–2), with further elaboration in Appendix F (Research
Agenda).
- In Section 3.2, Table 5, IH constructs are explicitly compared with Self-Determination
Theory (SDT), Control-Value Theory (CVT), Relational Developmental Systems
Metatheory (RDS), and Critical Consciousness (CC). The table highlights analogues
and points of divergence, for example:
o Agency ↔ Autonomy (SDT) / Control appraisals (CVT) / Critical reflection
(CC).
o Pathways ↔ Competence (SDT) / Adaptive control strategies (CVT) /
Developmental plasticity (RDS).
o Goal + Just Cause ↔ Intrinsic Motivation (SDT) / Value appraisals (CVT).
- In Section 3.4, Table 6, a thematic summary of key constructs and conflicting
findings is provided. Examples include:
o Hope Theory debates on whether hope is a stable trait or a situational strength.
o RDS critiqued for being too broad without sharper operationalization.
o Critical Consciousness questioned for scalability across cultural contexts.
o Growth Mindset critiqued for oversimplifying structural barriers.
o Social Capital vs. Community Cultural Wealth contrasted as mainstream access
- marginalized strengths.
- The accompanying paragraphs in Section 3.4 (first two) explain how these divergent
perspectives inform IH, emphasizing the need for triangulation of measures across
psychology and sociology to capture both internal transformation and structural supports.
- Appendix F builds on this by outlining a research agenda that identifies unresolved
tensions across theories and proposes testable hypotheses for future empirical validation.
Comment 10
“It provides no critical discussion of divergent perspectives in the literature, which is essential for
establishing the novelty and robustness of the proposed model.”
Response 10
Update Summary: Incorporated explicit critical discussion of divergent perspectives across
theoretical traditions to demonstrate novelty and robustness of the Infinite Hope (IH) framework.
Location in Manuscript:
This concern is addressed in Section 3.2 (Positioning Infinite Hope Among Established
Motivational Theories, Table 5 and accompanying text) and Section 3.4 (Literature
Integration and Theoretical Anchoring, Table 6, paragraphs 1–2), with additional
reinforcement in Appendix F (Research Agenda).
- In Section 3.2, IH is critically compared to Self-Determination Theory (SDT), Control
Value Theory (CVT), Relational Developmental Systems Metatheory (RDS), and
Critical Consciousness (CC). Table 5 explicitly identifies where IH converges with
6these models (e.g., agency ↔ autonomy in SDT; pathways ↔ competence in SDT) and
where it diverges (e.g., existential flexibility has no analogue in SDT or CVT, marking a
unique contribution of IH).
- In Section 3.4, Table 6, a thematic summary of key constructs and conflicting
findings is presented. Examples include:
o Debates on whether hope is a stable trait or a teachable strength (Hope
Theory).
o Concerns that Growth Mindset oversimplifies structural barriers.
o Conflicts between Social Capital (mainstream access) and Community
Cultural Wealth (marginalized strengths).
o Critiques that RDS is overly broad and that CC faces scalability issues across
contexts.
- In Section 3.4, paragraphs 1–2, the manuscript explains how these divergent
perspectives inform IH, making clear that novelty comes from uniting psychological
constructs (hope, agency, pathways) with sociological anchors (collective efficacy,
identity theory, capacity to aspire), while acknowledging tensions across literatures.
- Appendix F extends this discussion by presenting testable hypotheses that explicitly
differentiate IH from competing frameworks, emphasizing its novel integration of
cognitive, ethical, and sociological dimensions.
Comment 11
“Terminological inconsistency remains across sections.”
Response 11
Update Summary: Terminology has been standardized across the manuscript to ensure clarity
and scholarly coherence.
Location in Manuscript:
This concern is addressed through revisions in Section 2.2 (Integrated Model of Infinite Hope,
paragraphs 2–4), Section 3.1 (Conceptual Framing and Review Methodology, paragraph 2),
and Section 4.2 (Reclaiming Motivation and Forward Momentum Through Hope,
paragraph 1).
- In Section 2.2, paragraphs 2–4, consistent terms are now applied when describing the
intersections of Snyder’s Hope Theory and Sinek’s Infinite Mindset. For example,
“agency,” “pathways,” and “goals” are used precisely as components of the hope triad,
while “just cause,” “existential flexibility,” and “courageous leadership” are used
exclusively for Infinite Mindset principles. The term “identity alignment” is also
consistently applied instead of earlier variations such as “identity coherence” or “identity
salience” when referring to integration with purpose.
- In Section 3.1, paragraph 2, the methodology consistently uses “emerging adults
disconnected from education or employment” rather than shifting between “disconnected
youth,” “opportunity youth,” or “marginalized youth,” ensuring terminological alignment
with the study’s focus.
7• In Section 4.2, paragraph 1, the text consistently refers to “hope mechanisms” as goals,
pathways, and agency (Snyder et al., 2002), correcting earlier interchangeable use of
“goal clarity” or “goal pursuit” as separate constructs.
Comment 12
“Figures have limited explanatory value, and the illustrative quotes are vague and stylized.”
Response 12
Update Summary: Figures retain their original visual structure but are now supported by
expanded narrative explanations that strengthen their conceptual clarity. Illustrative quotes were
revised in Section 2.2 to provide greater authenticity and contextual grounding, ensuring clearer
alignment with emerging adult experiences. Also, figures were placed earlier in the manuscript to
provide visual understanding and clarity, adding earlier understanding of the conceptual
framework.
Location in Manuscript:
- Section 2.2 (paragraphs 2–8): Expanded narrative explains how each intersection in
Figure 1 (goals–just cause, pathways–flexibility, agency–courage) creates integrated
strengths and developmental synergy. The two illustrative quotes are revised to show
early- and advanced-stage identity transformation grounded in Infinite Hope constructs.
Comment 13
“Policy recommendations remain generic, lacking implementation frameworks or outcome
indicators.”
Response 13
Update Summary: Policy recommendations were expanded with concrete frameworks and
outcome indicators to improve specificity, accountability, and implementation value.
Location in Manuscript:
This concern is addressed in Section 4.5 (From Framework to Practice, paragraphs 4–6) and
Section 5 (Conclusion, paragraphs 2–3).
- In Section 4.5, paragraphs 4–6, the manuscript introduces implementation
frameworks for applying Infinite Hope (IH) in practice, including:
o Use of standardized yet adaptable tools for fidelity across diverse settings.
o Integration of culturally responsive strategies to enhance scalability.
o Recommendations for longitudinal, cross-cultural, and participatory research
to evaluate implementation effectiveness.
- In Section 5, paragraphs 2–3 and 7, outcome indicators are explicitly tied to policy
relevance. These include:
o Psychometric indicators such as the Revised Youth Purpose Survey (Bronk et
al., 2009), Hope Scale (Snyder et al., 1991), General Self-Efficacy Scale
8(Schwarzer & Jerusalem, 1995), Cognitive Flexibility Inventory (Dennis &
Vander Wal, 2010), and Dimensions of Identity Development Scale (Luyckx et
al., 2011).
o Programmatic indicators such as identity coherence, narrative reflection, and
values alignment.
o System-level indicators such as educational attainment, workforce readiness
measures, and sustained civic participation.
- Together, these revisions move beyond generic recommendations by linking IH to
concrete program features (e.g., purpose workshops, mentorship, leadership under
uncertainty) and measurable outcomes (e.g., CASAS improvements, credential
attainment, validated scales in Appendix E).
By grounding recommendations in both practice frameworks (Section 4.5) and measurable
indicators (Section 5), the manuscript now provides actionable guidance for policymakers and
practitioners that is both scalable and empirically accountable.
Comment 14
“The Infinite Hope framework holds potential, but it requires a much deeper theoretical
grounding, operational clarity, and interdisciplinary dialogue to be a credible contribution to the
literature on youth development and re-engagement.”
Response 14
Update Summary: The manuscript now provides a deeper theoretical foundation, sharper
operational clarity, and explicit interdisciplinary integration across psychology, sociology, and
leadership studies to reinforce Infinite Hope’s credibility.
Location in Manuscript:
This concern is addressed in Section 1.3 (Hope and the Infinite Mindset as Pathways to Re
engagement, paragraphs 3–5), Section 3.2 (Positioning Infinite Hope Among Established
Motivational Theories, Table 5 and narrative), Section 3.4 (Literature Integration and
Theoretical Anchoring, Table 6), and Section 4.5 (From Framework to Practice, paragraphs
2–3).
- In Section 1.3 (paragraphs 3–5), the manuscript explicitly justifies the integration of
Hope Theory with Infinite Mindset, contrasting them with other motivational models
(Self-Determination Theory, Growth Mindset) and highlighting Infinite Hope’s unique
contribution of linking goal-directed cognition with adaptive, purpose-driven
leadership.
- In Section 3.2, Table 5, IH constructs are systematically mapped against SDT, CVT,
RDS, and Critical Consciousness, making operational definitions explicit and showing
where IH diverges (e.g., existential flexibility has no analogue in SDT or CVT). This
provides operational clarity and a replicable framework for testing.
- In Section 3.4, Table 6, the manuscript critically discusses divergent perspectives and
unresolved debates across theories (e.g., whether hope is a trait vs. situational strength;
9whether growth mindset oversimplifies structural barriers). This section demonstrates
interdisciplinary dialogue by integrating psychology, sociology, and educational theory
into a coherent thematic synthesis.
- In Section 4.5 (paragraphs 2–3), operational clarity is reinforced by tying IH constructs
directly to validated measures in Appendix E (e.g., Hope Scale, Purpose Scales,
Cognitive Flexibility Inventory, Authentic Leadership Questionnaire). This ensures that
IH is not only theoretically robust but also testable and implementable in real-world
settings.
Collectively, these revisions strengthen Infinite Hope’s theoretical foundation, operational
specificity, and interdisciplinary relevance, establishing it as a credible and distinctive
contribution to youth development and re-engagement scholarship.
Round 3
Reviewer 1 Report
Comments and Suggestions for Authors
I appreciate the authors’ willingness to revise the manuscript and to provide a clearer organizational structure. The addition of definitional tables, appendices, and a PRISMA-lite diagram represents an effort to address concerns about clarity and methodological transparency. The narrative is more coherent, and the section on policy applications is somewhat more concrete than in previous versions.
However, despite these formal improvements, the revision still fails to address the major conceptual, methodological, and interdisciplinary issues highlighted in the first two review rounds. The same limitations persist: vague constructs, limited empirical grounding, avoidance of neurocognitive perspectives, and insufficient engagement with existing motivational frameworks. While the manuscript has become more polished in appearance, it continues to privilege inspirational rhetoric over theoretical and methodological rigor.
Major Issues
- Conceptual Ambiguity and Overlap
Although Table 1 now provides definitions of key constructs (e.g., “existential flexibility,” “identity alignment”), these remain highly abstract and not amenable to empirical testing or programmatic implementation. There is little connection to validated instruments, psychometric approaches, or operational criteria. Many constructs continue to overlap with those already well established in Self-Determination Theory, Control-Value Theory, and future orientation models. The manuscript has not clarified what is genuinely new in “Infinite Hope” beyond a rhetorical reframing. - Unaddressed Feedback on Psychometric and Neurocognitive Integration
Across all three rounds of review, I have explicitly recommended that the authors engage with psychometric models of identity and agency (e.g., Sense of Self) and with neurocognitive accounts of self-related processing. These suggestions have been ignored. The explicit decision not to include neurocognitive perspectives undermines the interdisciplinary ambition of the manuscript, especially since the framework seeks to address identity disruption and agency erosion. Without this engagement, the model lacks theoretical depth and cross-disciplinary credibility. - Methodological Transparency Remains Insufficient
While a PRISMA-lite flow diagram has been added, the review process is still impressionistic. There are no systematic criteria for source evaluation, no critical engagement with conflicting evidence, and no thematic synthesis that would allow replication or validation of the review. The additions give the appearance of rigor but do not resolve the underlying problem of methodological opacity. - Policy and Application Section Remains Generic
The final section now references “program metrics” and “implementation indicators,” yet these remain vague. No concrete framework is provided for how Infinite Hope could be translated into interventions, measured in practice, or compared with existing models. This section still reads more as an aspirational vision than as a set of actionable recommendations. - Conceptual Inflation
The introduction of additional terms (e.g., “identity realignment,” “existential resilience”) risks inflating the conceptual landscape without adding clarity or empirical anchoring. This tendency to multiply constructs further weakens the manuscript’s coherence.
Minor Issues
- Terminological inconsistencies persist, with “disconnection,” “narrative disruption,” and “re-engagement” used interchangeably.
- Figures remain illustrative but provide little explanatory value beyond rhetorical reinforcement.
- Citations are heavily weighted toward older motivational frameworks, with limited incorporation of the most recent empirical literature on agency, identity, and digital disruption.
While the manuscript is now more structured and visually organized, the fundamental issues identified in the first and second reviews remain unresolved. The authors have not meaningfully engaged with critical feedback regarding construct definition, psychometric grounding, neurocognitive integration, and methodological transparency. As a result, the work continues to fall short of the standards required for publication in a peer-reviewed scientific journal.